# Were early Archean carbonate factories major carbon sinks on the juvenile Earth?

Xiang, Wanli[1, 2], Duda, Jan-Peter[2], Pack, Andreas[3], van Zuilen, Mark[4], Reitner, Joachim[2, 5]

[1] College of Tourism and Geographical Science, Leshan Normal University, Leshan, 614000, China.

[2] Department of Geobiology, University of Göttingen, Göttingen, 37077, Germany.

[3] Department of Geochemistry and Isotope Geology, University of Göttingen, Göttingen, 37077, Germany.

[4] CNRS-UMR6538 Laboratoire Geo-Ocean, Institut Universitaire Européen de la Mer (IUEM), Université de Bretagne Occidentale, Plouzané, 29280, France.

[5] Göttingen Academy of Science and Humanities in Lower Saxony, Göttingen, 37073, Germany.

*Correspondence to*: Reitner Joachim (jreitne@gwdg.de)

**Abstract.** Paleoarchean carbonates in the Pilbara Craton (Western Australia) are important archives for life and environment on early Earth. Amongst others, carbonates occur in interstitial spaces of ca. 3.5–3.4 Ga pillow basalts (North Star-, Mount Ada-, Apex-, and Euro Basalt, Dresser Formation) and associated with bedded deposits (Dresser- and Strelley Pool Formation, Euro Basalt). This study aims to understand the formation and geobiological significance of those early Archean carbonates by investigating their temporal-spatial distribution, petrography, mineralogy, and geochemistry (e.g., trace elemental compositions, $\delta^{13}C$, $\delta^{18}O$). Three carbonate factories are recognized: (i) an oceanic crust factory, (ii) an organo-carbonate factory, and (iii) a microbial carbonate factory. The oceanic crust factory is characterized by carbonates formed in void spaces of basalt pillows (referred to as "interstitial carbonates" in this work). These carbonates precipitated inorganically on and within the basaltic oceanic crust from $CO_2$-enriched seawater and seawater-derived alkaline hydrothermal fluids. The organo-carbonate factory is characterized by carbonate precipitates that are spatially associated with organic matter. The close association with organic matter suggests that the carbonates formed via organo-mineralization, that is, linked to organic macromolecules (either biotic or abiotic) which provided nucleation sites for carbonate crystal growth. Organo-carbonate associations occur in a wide variety of hydrothermally influenced settings, ranging from shallow marine environments to terrestrial hydrothermal ponds. The microbial carbonate factory includes carbonate precipitates formed through mineralization of extracellular polymeric substances (EPS) associated with microbial mats and biofilms. It is commonly linked to shallow subaquatic environments, where (anoxygenic) photoautotrophs might have been involved in carbonate formation. In case of all three carbonates factories, hydrothermal fluids seem to play a key-role in the formation and preservation of mineral precipitates. For instance, alkaline earth metals and organic materials delivered by fluids may promote carbonate precipitation, whilst soluble silica in the fluids drives early chert formation, delicately preserving authigenic carbonate precipitates and

associated features. Regardless of the formation pathway, Paleoarchean carbonates might have been major carbon sinks on the early Earth, as additionally suggested by carbon isotope mass balances indicating a carbon flux of 0.76-6.5 × $10^{12}$ mol/year. Accordingly, these carbonates may have played an important role in modulating the carbon cycle and, hence, climate variability, on the early Earth.

## 1 Introduction

Biogeochemical carbon cycling plays a crucial role in maintaining the stability of modern Earth's climate system (Ciais et al., 2013). It encompasses carbon fluxes between sources (e.g., volcanic and metamorphic $CO_2$) and various sinks (e.g., terrestrial silicate weathering, seafloor carbonatization, biomass build-up, deposition of carbonates and organic matter, carbon subduction into the mantle) (Ciais et al., 2013; Suarez et al., 2019). In case of carbon sinks, carbonates and organic matter stand out as the two paramount reservoirs (Gislason and Oelkers, 2014; Hoefs, 2018; Shields, 2019; Canfield, 2021). Out of these two reservoirs, carbonate rocks are particularly significant, serving as a primary carbon sink over geological timescales (Veizer et al., 1982; Nakamura and Kato, 2004; Canfield, 2021). Understanding the formation and evolution of carbonate factories— that is, conceptual models encompassing carbonate production and associated processes at various scales, from local precipitation to global sedimentation (Schlager, 2000; Schrag et al., 2013; Reijmer, 2021)—is therefore essential for comprehending the dynamics of the carbon cycle and its implications for climate change.

Throughout most of Earth's history, carbonate precipitation has been closely linked to biological processes, ranging from direct to indirect precipitation (that is, biologically controlled vs induced) (Flügel, 2010). Over the past couple of years, awareness has risen that carbonate precipitation can also be induced by organic matter (i.e., "organo-mineralization"), regardless of its origin (Addadi and Weiner, 1985; Reitner, 1993; Reitner et al., 1995a, b, 2000, 2001; Trichet and Défarge, 1995; Pei et al., 2021, 2022). Based on previous works about "cold-water carbonates" (Lees and Buller, 1972) and "mud-mound carbonates" (Reitner and Neuweiler, 1995), Schlager (2000) summarized three modern carbonate factories in the marine benthic zone, namely tropical shallow-water system, cool and deep-water system, and mud-mound/microbial buildup system. Since then, the carbonate factory concept has been extended across spatial and temporal scales (Pomar and Hallock, 2008; Reijmer, 2021; Pei et al, 2021, 2022; Wang et al., 2023). Despite these years of intense research, carbonate factories are still poorly understood, particularly during the Archean eon (4.0 to 2.5 billion years ago, Ga), when life on Earth was just in its infancy. During this eon, which accounts for about one-third of our planet's history, the Earth system experienced significant geobiological developments, such as declining volcanic activity and decreasing surface temperatures (Lunine and Lunine, 1998; Nisbet and Sleep, 2001; Lowe and Tice, 2007; Sengupta et al., 2020; Lowe et al., 2020; Reinhardt et al., 2024). Hence, geological and biological key-processes – and by extension Archean carbonate factories – must have been very different as compared to any later stage in Earth's history.

One of the most important early Earth records is the ca. 4.0–3.6 Ga Isua Supracrustal Belt (ISB; West Greenland), albeit being highly metamorphic (amphibolite facies: Nutman et al., 2019a, b). Rocks of the 3.5–3.2 Ga East Pilbara Terrane (EPT; Pilbara Craton, Western Australia) and the 3.6–3.2 Ga Barberton Greenstone Belt (BGB; South Africa) in contrast are well preserved in terms of metamorphism (prehnite-pumpellyite to greenschist facies, i.e. max. 250–300 °C: Van Kranendonk et al., 2019a; Hickman-Lewis et al., 2019) and hence provide valuable windows into early Archean carbon sinks. Indeed, rocks of the EPT are for instance well known to contain carbonates associated with microbial facies (i.e. stromatolites; Van Kranendonk, 2006, 2007; Allwood et al., 2006, 2007; Wacey, 2010; Lepot et al., 2013; Sugitani et al., 2015a) and pillowed basalts (Kitajima et al., 2001; Nakamura and Kato, 2002, 2004; Terabayashi et al., 2003; Marien et al., 2023) as well as carbonaceous organic matter (primarily preserved in chert and carbonate: Marshall et al., 2007; Bontognali et al, 2012; Duda et al., 2016, 2018; Flannery et al., 2018; Weimann et al., 2024; Mißbach et al., 2021). With regard to carbonates, most studies have focused on occurrences associated with microbial facies in the ~3.4 Ga Strelley Pool Formation, yet those constitute a minor component within the EPT lithostratigraphy (Van Kranendonk et al., 2007b).

Basaltic rocks of the EPT show evidence of pervasive carbonatization and silicification associated with hydrothermal processes in subaquatic environments (Kitajima et al., 2001; Nakamura and Kato, 2002, 2004; Terabayashi et al., 2003). Carbon flux estimates based on observations in this region suggest that hydrothermal carbonatization of pillowed seafloor basalts constituted a significant $CO_2$ sink in the early Archean (Nakamura and Kato, 2004). However, these estimates did not consider interstitial carbonates — that is, carbonates that precipitated in void spaces between pillow basaltic rocks (Marien et al. 2023; this work). Over the past few decades, the significance of carbon sequestration through hydrothermal alteration of oceanic crust has become obvious, underscoring the meaning of these processes for the global carbon cycle and, by extension, long-range climate regulation (Alt and Teagle, 1999; Bach et al., 2011; Coogan and Gillis, 2013; Krissansen-Totton et al., 2015, 2018). Indeed, more recent mass balance models for total organic carbon burial ($f_{org}$) take also authigenic carbonates, including those formed through oceanic crust carbonatization, into account (e.g., Krissansen-Totton et al. 2015). Nevertheless, the informative value of these approaches is limited due to the scarcity of theoretical frameworks and geological baseline data that would allow to constrain such additional carbon sinks, thereby impeding a comprehensive understanding of carbon cycle dynamics over geological time scales.

To address this issue, this study investigates early Archean carbonates in the EPT, including interstitial carbonates associated with basalts, carbonate stromatolites and other sedimentary carbonates. The combination of detailed petrography with mineralogical and geochemical analyses (e.g., trace elemental compositions, $\delta^{13}C$, $\delta^{18}O$) provides novel insights into the formation of carbonates during the early Archean. Based on this information, the fraction of carbon in the ocean being sequestered as inorganic and organic carbon was calculated and discussed with respect to overall carbon flux dynamics. The results of this study demonstrate the presence of various types of carbonate factories on the juvenile Earth and indicate that they might play a significant role in the early global carbon cycle and, hence, climate system.

## 2 Geological settings

The EPT (3.53–3.17 Ga) in Western Australia is famous for its well-preserved Paleoarchean volcano-sedimentary successions, which provide the world's most complete record of the evolution of the geo-, hydro-, bio- and atmosphere on the early Earth (Van Kranendonk et al., 2007a, b; Hickman and Van Kranendonk, 2012a, b). A particular interest is the Pilbara Supergroup, a 20 km thick succession of mainly volcanic rocks that can be subdivided into (from bottom to top) the Warrawoona Group (3.53–3.43 Ga), the Kelly Group (3.42–3.32 Ga), the Sulphur Springs Group (3.27–3.23 Ga), and the Soanesville Group (ca. 3.19 Ga) (Van Kranendonk et al., 2002, 2007b; Rasmussen et al., 2007; Hickman and Van Kranendonk, 2012a, b). The lower three groups comprise ultramafic to felsic volcanic rocks, chemical and clastic deposits, as well as swarms of subseafloor hydrothermal silica ± barite veins (Van Kranendonk, 2006). The tectonic setting of the EPT is controversial, ranging from mid-ocean ridge and island arc (Ueno et al., 2001; Komiya et al., 2002; Kato and Nakamura, 2003) to a thick ocean volcanic plateau (Smithies et al., 2003, 2005, 2007a, b; Van Kranendonk, 2006; Van Kranendonk et al., 2007a, b, 2019a).

A characteristic feature of the EPT is the so-called dome-and-keel structure, consisting of a central nucleus of the 3459 ± 18 Ma North Pole Monzogranite ("North Pole Dome") surrounded by little-deformed, predominantly mafic volcanic rocks of the Warrawoona Group and Kelly Group (Hickman and Van Kranendonk, 2012a) (Fig. 1). The oldest basaltic formation in this area is the North Star Basalt (3490 ±15 Ma Ar/Ar), which is overlain by the Dresser Formation (3481 ± 2 Ma U-Pb) consisting of chert ± barite beds and veins that are associated with pillowed basalts and dolerite (Van Kranendonk et al., 2008; Hickman and Van Kranendonk, 2012b). Atop the Dresser Formation follows (from base to top) a ~ 4 km thick succession of mafic volcanic rocks (Mount Ada Basalt), a < 1.3 km thick succession of felsic volcanic rocks (Duffer Formation, Panorama Formation), and a < 150 m thick package of jasper (Marble Bar Chert Member, Towers Formation) (Byerly et al., 2002; Hickman and Van Kranendonk, 2012b). In the eastern part of the dome, the Panorama Formation is underlain by the Apex basalt (Nakamura and Kato, 2004), which is dated to 3463–3454 Ma based on zircon U-Pb ages of the underlying Duffer Formation and the overlying Panorama Formation (Thorpe et al., 1992; McNaughton et al., 1993). Surrounding the central dome, the Panorama Formation is disconformably overlain by the Strelley Pool Formation (SPF, 3414 ± 34 Ma, U–Pb ages, Gardiner et al., 2019), which is known for its distinctive stromatolites (e.g., Lowe, 1980, 1983; Hofmann et al., 1999; Van Kranendonk et al., 2003; Allwood et al., 2006a; Hickman et al., 2011; Duda et al., 2016), followed by the high-Mg and tholeiitic Euro Basalt (3350 ± 3 – 3335 ± 7 Ma, GSWA, 2013) (Van Kranendonk et al., 2006; Hickman and Van Kranendonk, 2012b).

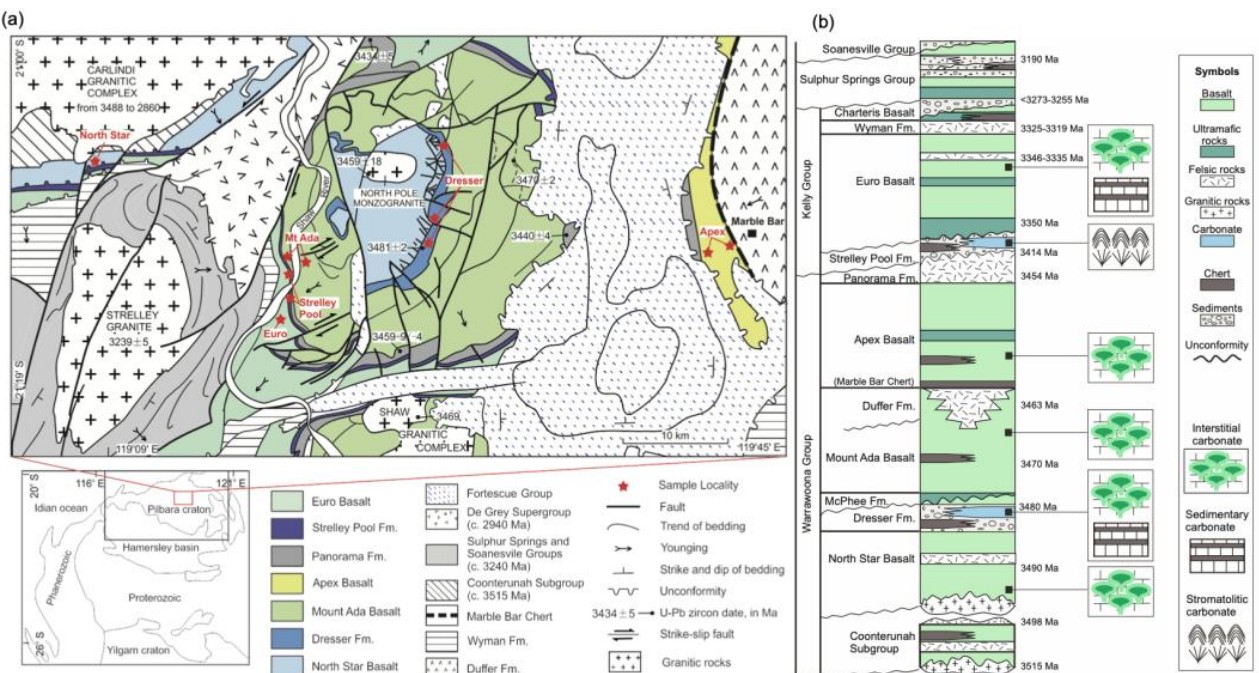

**Figure 1: (a) Simplified geological map of the North Pole Dome, Eastern Pilbara Terrane, Western Australia (adapted from Van Kranendonk and Hickman, 2000, Hickman and Van Kranendonk, 2012b) including sampling localities (red stars). (b) Simplified stratigraphy of the studied Archean rocks (adapted from Van Kranendonk et al., 2007b).**

## 3 Materials and methods

### 3.1 Sample locality

Paleoarchean carbonate rocks analyzed in this study derive from the North Pole Dome in the EPT (Fig. 1) and were collected from existing drill cores stored at the Geological Survey of Western Australia (Agouron Institute Drilling Project, AIDP) as well as during field campaigns organized by the German Research Foundation (DFG) Priority Program 1883 "Building a Habitable Earth" together with Australian colleagues. Interstitial carbonates were sampled from the ~3.49 Ga North Star Basalt (drill core 102 AIDP-1; 21°06'38"S, 119°06'4"E, French et al., 2015), the ~3.46 Ga Apex Basalt ("Schopf Locality" at Chinaman Creek; Schopf, 1993), the ~3.47 Ga Mount Ada Basalt and the ~3.35 Ga Euro Basalt (both near "Trendall Locality" at Shaw River; Hickmann et al., 2011), as well as from the Middle Basalt Member of ~3.48 Ga Dresser Formation (Dresser Barite Mine). Bedded sedimentary carbonates were sampled from the Dresser Formation at the "Tsunami Locality" (Runge et al. 2022) near the Dresser Barite Mine, and the Euro Basalt at the east side of the Shaw River near the "Trendall Locality". Stromatolitic carbonates were collected from the Strelley Pool Formation at the western side of Shaw River.

In order to better understand depositional environments of the studied EPT carbonates, we additionally analyzed the carbon and oxygen isotopic compositions of diverse reference materials for comparison. They include carbonate inclusions in black

barites from the Dresser Formation (drill cores PDP 2b and 2c), rhodochrosites in cherts from the ~3.25 Ga Fig Tree Group

(Heinrichs, 1980; Rincon-Thomas et al 2016), carbonates of debated origin in the vicinity of the controversial ~3.7 Ga stromatolite site in the ISB in Greenland (Nutman et al., 2016; Allwood et al., 2018; provided by van Zuilen, 2018), as well as carbonatites from ~540 Ma Fen Complex in Norway (Andersen and Taylor, 1988) and the ~16 Ma Kaiserstuhl Volcanic Complex in Germany (Kraml et al., 2006).

## 3.2 Methods

### 145 3.2.1 Petrography and geochemical imaging

Petrographic thin sections were prepared (polished to approximately 60 μm thickness) for all samples and examined using a Zeiss SteREO Discovery V12 stereomicroscope coupled with an AxioCam MRc camera. Selected carbonates were additionally analyzed with a Cathodoluminescence (CL) microscope. CL images were acquired with a Cambridge Instruments Citl CCL 8200 Mk3A cold-cathode system linked to a Zeiss Axiolab microscope (operating voltage of approximately 15 kV and electric

current of approximately 250-300 μA) and a Zeiss AxioCam 703 camera.

Minerals were identified by their optical characteristics and Raman spectroscopy, using a Horiba Jobin-Yvon LabRam-HR 800 UV spectrometer with a focal length of 800 mm and an excitation wavelength of 488 nm produced by an Argon ion laser (Melles Griot IMA 106020B0S) and with a WITec alpha300 R fibre-coupled ultra-high throughput spectrometer. The former spectrometer was calibrated using a silicon standard with a major peak at 520.4 $cm^{-1}$, and the spectra were processed using

software Fityk (Wojdyr, 2010) and comparatively analyzed based on references from the RRUFF database.

Element distributions were mapped using a Bruker M4 Tornado micro-X-ray fluorescence (μXRF) instrument equipped with a XFlash 430 Silicon Drift Detector. Measurements were performed at a voltage of 50 kV and a current of 400 μA with a spot size of 20 μm and a chamber pressure of 20 mbar.

### 3.2.2 Stable carbon and oxygen isotopes ($\delta^{13}C$, $\delta^{18}O$)

For stable isotope analyses, sample chips (diameter ~1 cm) were obtained from pristine areas (i.e., free of visible alteration, inclusions, and secondary porosity) using a microdrill. The sample chips were cleaned three times in ethanol using an ultrasonic bath and dried at room temperature before being crushed into small pieces. Carbonate was then picked out and powdered in an agate mortar and well homogenized. Additionally, some carbonate facies, including carbonate veinlets and carbonate inclusions, were extracted using a drill from individual mineral phases from polished rock slabs.

Carbon and oxygen stable isotopes of the carbonates were measured at 70 °C using a Thermo Scientific Kiel IV carbonate device coupled with a Finnigan DeltaPlus gas isotope mass spectrometer at the Geoscience Center of the Georg-August-Universität Göttingen. All results were normalized as delta values $\delta^{13}C_{carb}$ and $\delta^{18}O_{carb}$ relative to the Vienna PeeDee Belemnite (VPDB) reference standard. The standard deviation is better than 0.03 ‰ for $\delta^{13}C_{carb}$ and 0.05 ‰ for $\delta^{18}O_{carb}$, calculated by multiple measurements of the in-house carbonate standard Solnhofen.

**4 Results**

**4.1 Interstitial carbonates**

**4.1.1 Host basalts**

The host basalts are pillow-shaped, internally subdivided into more crystalline interiors and quenched glassy rims, and commonly locally cut by tectonic fractures (Fig. 2). The interspaces and fractures are filled with carbonate minerals and chert.
In most outcrops, the host basalts and interstitial carbonate minerals are weathered, resulting in orange to brownish colors.

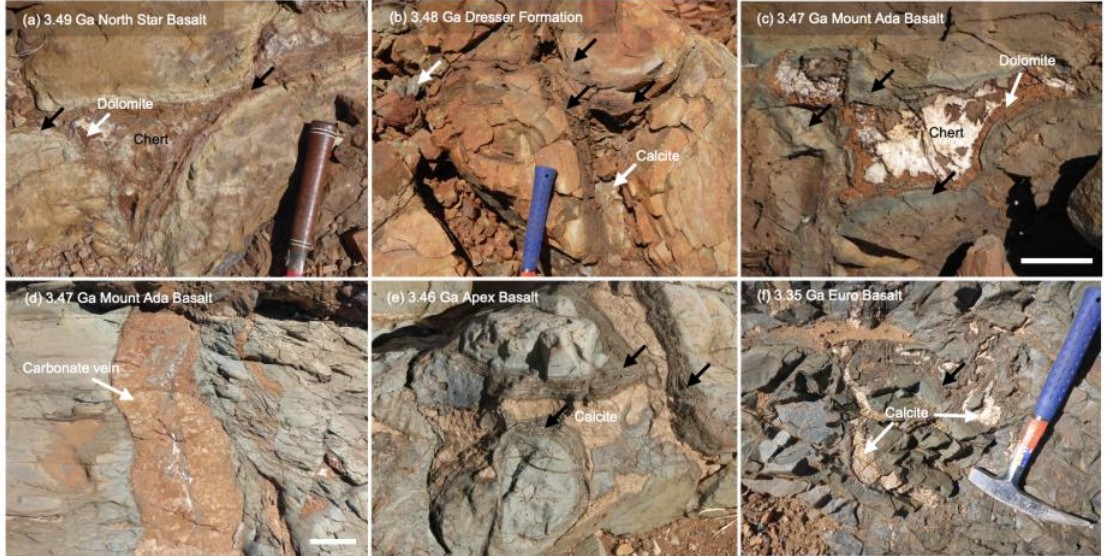

**Figure 2: Outcrop photos of Archean pillow basalts from the North Pole Dome, Eastern Pilbara Terrane, Western Australia. The pillows consist of relatively unaltered cores surrounded by quenched rims (black arrows), indicating a sub-aquatic formation. Interspaces between pillows are filled with carbonate minerals ("interstitial carbonate"; white arrows). (a) Outcrop of surficially**
**weathered 3.49 Ga North Star Basalt with interspaces filled by Fe-dolomite and chert cement; (b) Outcrop of surficially weathered 3.48 Ga Dresser Formation lacking interstitial calcite due to weathering; (c) Outcrop of surficially weathered 3.47 Ga Mount Ada Basalt with interspaces filled by fibrous isopachous Fe-dolomite (brown colours, due to weathering) and white chert. It is locally cut by deep carbonate veins shown in (d), implying later fluid circulation. (e) Outcrop of little weathered 3.46 Ga Apex Basalt with interspaces filled with pink calcite, basaltic breccia and minor chert. (f) Outcrop of little weathered ~3.35 Ga Euro Basalt with**
**interspaces and fractures filled by pink calcite. The lengths of the brown and blue hammers are ca. 30 cm and ca. 40 cm, respectively. Scale bars are 10 cm in (c) and 20 cm in (d), respectively.**

Although the host basalts show secondary mineral assemblages indicative of greenschist metamorphism (calcite + chlorite + anatase + quartz ± pyrite), phenocrysts (i.e., plagioclase and pyroxene) can still be recognized in the basalt interior of the well-preserved samples, e.g. A22 from the Apex Basalt (Figs. 3a, 4a). Notably, the well-preserved basalts exhibit concentric green
ophitic-holohyaline interiors and yellow-green quenched margins. In the margins, the size and density of ovoid spherulites and

variolites (amygdules) decrease outwards, merging into the glassy zone (Fig. S1a-c). Carbonate minerals are particularly prominent in voids, veins and variolites within alteration zones, as illustrated by μXRF element overlay images of Si, Ca and Mg (Fig. 3b) as well as by calculated Ca mass changes (Fig. S14b; Appendix C). Except for the devitrified volcanic glass, Si is rich in the interior of the pillow basalt but rare in the alteration zones (Figs. 3b, S14), implying a Si loss during basalt carbonatization. Si yielded during this process was likely enriched in fluids, resulting in chert cementation of interstitial carbonates (Fig. 4). The process can be summarized as follows (Eq.1; note that "CaSiO$_3$" refers to calcium silicate minerals):

$$\text{"CaSiO}_3\text{"}+ CO_2 + H_2O \rightarrow \text{"CaSiO}_3\text{"} + H_2CO_3 \rightarrow CaCO_3 + SiO_2 + H_2O \qquad (Eq.\,1)$$

In case of the altered host basalts, progressive deformation and later stage metamorphism are evidenced by the migration and breakup of secondary minerals (e.g. chlorite), erased volcanic textures, as well as by the presence of schistose areas (Figs. 5a, S1d-i). The migration of chlorite, which is a dominant Fe-bearing secondary mineral, caused a loss of Fe in weathered basalts (Fig. 5a). Minerals of the chlorite-group frequently occur in interstitial carbonates close to, and within, tectonic fractures in the pillow basalts (Fig. 5b).

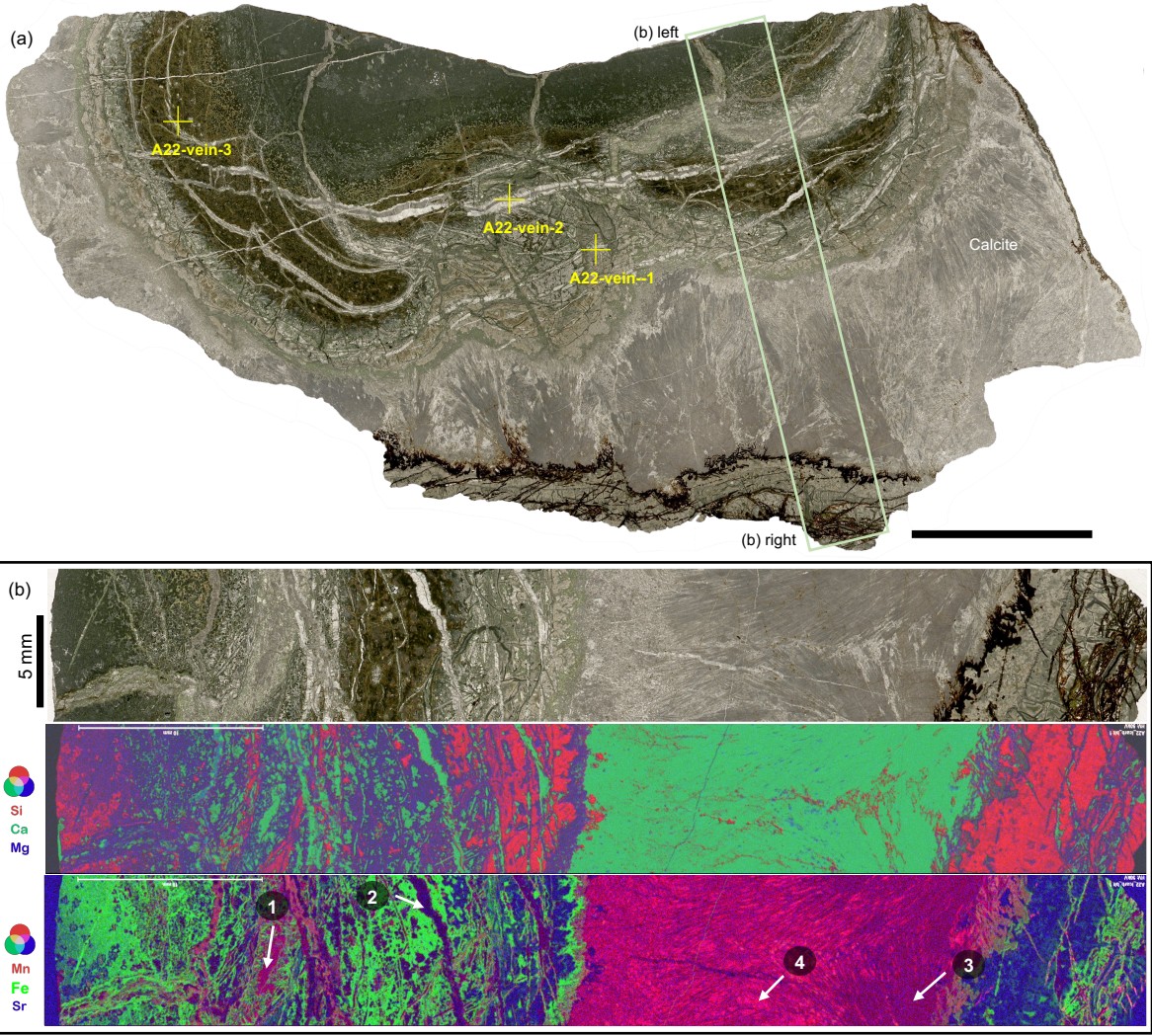

**Figure 3: (a) Thin section scan image (transmitted light) of sample A22 from the Apex Basalt, showing concentric pillow structures of the basalt and well-preserved primary acicular crystal-fans of interstitial carbonates (mainly calcite). Yellow crosses mark the positions of subsamples analysed for stable oxygen and carbon isotopes (Table 1, Fig. S3). The scale bar in (a) is 20 mm. (b) Blow-up image and μXRF mappings of rectangle area highlighted in (a). The false-colour overlapping image of Si (red), Ca (green) and Mg (blue) in the middle panel is well in line with interspaces dominantly filled by calcite with minor chert. In addition, the quenched margin of the basalt seems to be relatively depleted in Si as compared to the core, implying a loss of Si during carbonatization processes (see Eq. 1). The Si yielded by carbonatization was likely enriched in fluids and resulted in the later cementation of interstitial calcite by chert. The false-colour overlapping image of Mn (red), Fe (green) and Sr (blue) in the lower panel highlights the presence of four calcite facies, that is, Mn-enriched syngenetic veins (white arrow 1), Mn-depleted later veins (white arrow 2), Mn-depleted acicular interstitial calcite (white arrow 3), and Mn-enriched calcite cement (white arrow 4). The images of each element are shown in Fig. S5.**

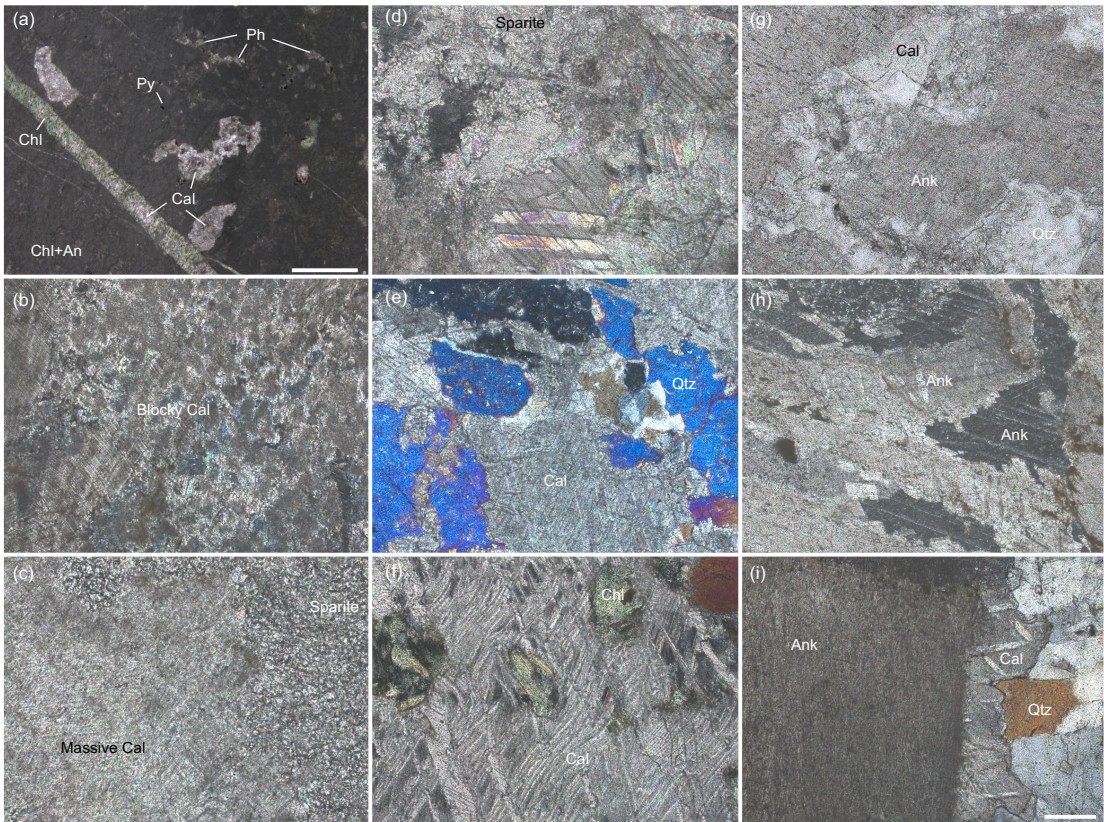

**Figure 4: Thin section photographs of altered interstitial carbonates from the North Pole Dome, Eastern Pilbara Terrane, Western Australia. (a) Phenocrysts can be recognized in the well-preserved host basalt, although the secondary mineral assemblage is indicative of greenschist metamorphism (calcite + chlorite + anatase + quartz ± pyrite). The acicular crystal-fan calcite is altered to blocky calcite (b) as well as to massive and sparitic calcite (c). (d) Large sparitic crystals in a wide fracture. (e) The blocky calcites**

**are cemented by quartz. (f) Metamorphic S-C fabrics of sparite and chlorite crystals indicate dynamic metamorphism. (g) Blocky ankerites often show calcite overgrowths at their edges. (h) Some ankerites exhibit features formed by recrystallization and neomorphism. (i) Along dewatering cracks, ankerites in deep carbonate veins are commonly overgrown by calcite and chert cement. (a) to (c): Apex Basalt; (d): Dresser Formation; (e) and (f): Euro Basalt; (g) to (i): Mount Ada Basalt. All photos except (g) were taken under cross-polarized light. Scale bar in (i) corresponds to 200 μm and is applicable to all photographs. Abbreviations: Ph-**

**phenocryst, Cal-calcite, Chl- chlorite, Qtz- quartz, Ank-ankerite.**

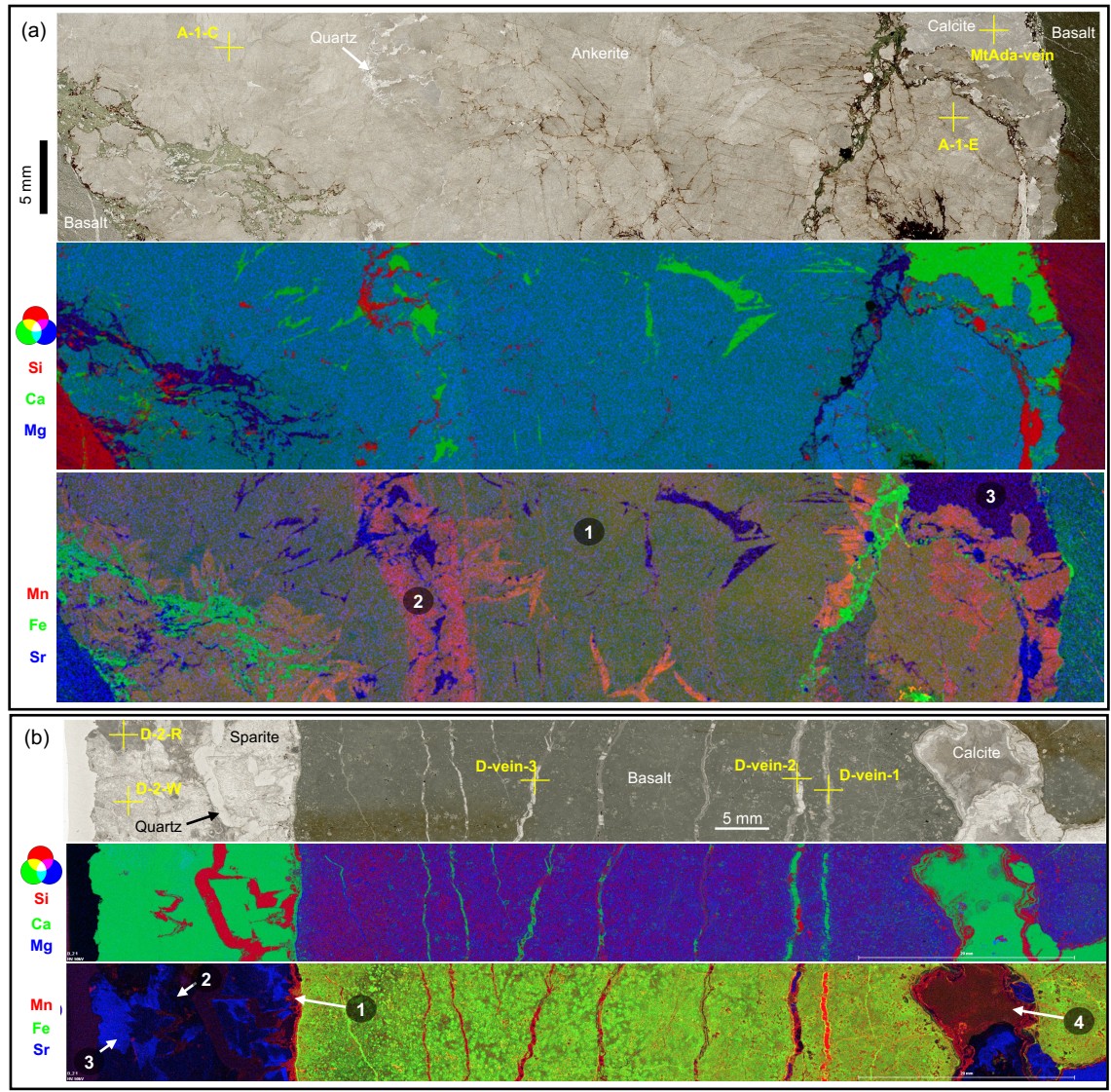

**Figure 5: Thin section scan images (transmitted light) and false-colour overlapping images of elements of interstitial carbonates in (a) the Mount Ada Basalt and (b) the Dresser Formation. (a) Interstitial carbonate in the Mount Ada Basalt consists of blocky and massive ankerite with minor calcite overgrowth and infilling quartz, as evidenced by Fig. S4b and Si (red), Ca (green) and Mg (blue) distributions shown in false-colour overlapping images (middle panel in a). False-colour overlapping images of Mn (red), Fe (green) and Sr (blue) (lower panel in a) highlight the presence of Mn-enriched ankerite (close to metabasalt; number 2 in the figure), Mn-depleted calcite (in later vein; number 3), and ankerite with intermediate enrichments of Mn (distant to metabasalt; number 1). Mn-enriched ankerite might be influenced by later fluids as indicated by calcite and quartz. (b) Interstitial carbonates in the Dresser Formation includes precipitates enriched in Mn (first and fourth generations, shown in the figure by the numbers and arrows), depleted in Mn and Sr (second generation), as well as Mn-depleted but Sr-enriched precipitates (third generation). Precipitates of**

the first and fourth generation seem identical to calcite occurring within parallel fractures of basalts, implying precipitation from similar fluids that derived from fluid-basalt reactions. The second and third generation are distinctive from the aforementioned generations, indicating a different origin of the later fluids. Yellow crosses mark the positions of subsamples analysed for stable oxygen and carbon isotopes (Table 1, Fig. S3). The images of each element are shown in Figs. S6, S7.

### 4.1.2 Primary carbonate phases

The primary mineral phase of interstitial carbonates is calcite. Acicular crystal-fans of calcite are only preserved in some samples from the Apex Basalt, but are reduced or absent in most other cases. The terminal tips of the acicular crystal-fans are partly recrystallized to sparitic calcite crystals (Fig. 4b). Microcrystalline ankerite is rarely observed at the basalt margin, mixing with microcrystalline quartz, chlorite, and anatase particles (~nm) (Figs. S2a, S4a). Minor chert locally infills the intercrystalline space of sparite and crystal-fan calcite. Primary carbonates occurring within basalts include blocky calcite in concentric syngenetic veins and fibrous isopachous calcite in tectonic fractures, often showing shear bending through dynamic crystallization (Fig. S2b, e).

Calcite is the dominant primary carbonate phase in all samples, which is in line with the spatially independent distributions of Ca, Mg and Si in the precipitates (Fig. 3b). However, distinct calcite phases show different contents of Mn, with acicular crystal-fan and fibrous isopachous precipitates being depleted in Mn relative to the associated intercrystalline calcites. Fe, Si and Mg are pervasive in basalt and fractures, which is due to the presence of chlorite group minerals (Fig. 5a).

### 4.1.3 Secondary carbonate facies

In many cases, primary interstitial calcite was affected by post-depositional alteration processes such as recrystallization and ankeritization. Recrystallization is widespread, involving the transformation of acicular crystal-fan calcite to inequigranular, blocky, massive, and sparry calcite (Fig. 4b–d). The recrystallized interstitial calcite is commonly cemented by quartz (Fig. 4e, g). In some samples, sparite exhibits a S-C fabric (Fig. 4f), indicative of deformation through dynamic metamorphism in a shear zone (Lister and Snoke, 1984). Noteworthy, sparite in sample D-2 from the Dresser Formation is rather associated with tectonic fractures than with basalt interspaces (Figs. 4d, S2 d-f; Xiang, 2023); therefore, it will be addressed as "fracture-filling calcite" in the following.

Carbonates from the Mount Ada Basalt underwent significant ankeritization, as indicated by abundant blocky and massive ankerite cemented by quartz (Fig. 4g). Rarely observed relict structures "floating" in ankerite evidence acicular crystal-fan calcite as precursor (Fig. S2c). The interstitial ankerite locally underwent recrystallization and neomorphism (Fig. 4h). Calcite veins locally cut the interstitial ankerite and the host basalt. Ankerite precipitates in those samples are commonly overgrown by calcite (Fig. 4g, i).

The secondary carbonates are either Mn- or Sr-enriched (Fig. 5), indicating the influence of at least two diagenetic fluids during later alteration. Secondary Mn-enriched carbonates include recrystallized interstitial calcites and ankerites as well as calcite cements within basalt fractures. Notably, the degree of Mn-enrichment in interstitial ankerites varies, with those formed

through recrystallization and neomorphism or closer to the basaltic parts being relatively more enriched (Fig. 5a). At the same time, calcites overgrowing interstitial ankerites and, even more so, within fractures are enriched in Sr (Fig. 5).

### 4.1.4 $\delta^{13}$C and $\delta^{18}$O values of interstitial carbonates

The interstitial carbonates (including both, calcite and ankerite) show $\delta^{13}$C values ranging from -2.37 to +0.99 ‰ (mean = 0.22 ± 0.98 ‰) and $\delta^{18}$O values ranging from -19.81 to -14.34 ‰ (mean = -17.57 ± 1.51 ‰) (Table 1). The fracture-filling calcites exhibit $\delta^{13}$C values ranging from 2.03 to 2.34 ‰ (mean = 2.19 ± 0.13 ‰) and $\delta^{18}$O values ranging from -17.91 to -13.03 ‰ (mean = -15.70 ± 2.53 ‰) (Table 1). Carbonates in veins (see Figs. 3, 5) have the slightly lower $\delta^{13}$C and $\delta^{18}$O values than interstitial carbonates in the same samples (Table 1, Fig. S3).

**Table 1: Stable carbon and oxygen isotopic compositions of the early Archean carbonates**

| Lithology | Mineralogy | Formation | Age (Ma) | SampleID | $\delta^{13}C_{VPDB}$ (‰) | s.d. | $\delta^{18}O_{VSMOW}$ (‰) | s.d. | $\delta^{18}O_{VPDB}$ (‰) |
|---|---|---|---|---|---|---|---|---|---|
| Interstitial Carb. | Calcite | Euro Basalt | 3350 | E-1 | 0.21 | 0.03 | 11.08 | 0.05 | -19.23 |
| | | | | E-2 | 0.99 | 0.03 | 10.78 | 0.05 | -19.52 |
| | | | | E-3 | -2.37 | 0.03 | 11.00 | 0.05 | -19.31 |
| | | Apex Basalt | 3460 | A14673-1 | 0.62 | 0.03 | 13.65 | 0.05 | -16.74 |
| | | | | A22-1 | 0.44 | 0.03 | 13.09 | 0.05 | -17.29 |
| | | | | A22-2 | 0.69 | 0.03 | 13.67 | 0.05 | -16.72 |
| | | | | ABAS-1 | 0.65 | 0.03 | 13.41 | 0.05 | -16.97 |
| | | | | ABAS-1 | 0.77 | 0.03 | 14.63 | 0.05 | -15.79 |
| | | | | Apex-1 | 0.25 | 0.03 | 12.79 | 0.05 | -17.58 |
| | | | | Apex-2 | 0.04 | 0.03 | 12.66 | 0.05 | -17.70 |
| | | | | Apex-3 | 0.21 | 0.03 | 13.00 | 0.05 | -17.37 |
| | Ankerite, calcite | Mt.Ada Basalt | 3470 | MtAda-1-C | 0.83 | 0.00 | 12.21 | 0.03 | -18.14 |
| | | | | MtAda-1-E | 0.77 | 0.03 | 11.77 | 0.05 | -18.57 |
| | | | | MtAda-2 | 0.52 | 0.03 | 14.36 | 0.03 | -16.05 |
| | Calcite | Dresser Fm. | 3480 | D-1 | 0.97 | 0.03 | 14.46 | 0.05 | -15.95 |
| | | | | D-3 | 0.63 | 0.03 | 10.49 | 0.05 | -19.81 |
| | | North Star Basalt | 3490 | CP-1 | -2.31 | 0.03 | 11.14 | 0.05 | -19.17 |
| | | | | CP-2 | 0.01 | 0.03 | 16.12 | 0.05 | -14.34 |
| Fracture Carb. | Calcite | Dresser Fm. | 3480 | D-2-IC-1 | 2.03 | 0.03 | 12.45 | 0.03 | -17.91 |
| | | | | D-2-IC-2 | 2.17 | 0.03 | 16.43 | 0.03 | -14.04 |
| | | | | D-2-R | 2.34 | 0.03 | 12.54 | 0.05 | -17.81 |
| | | | | D-2-W | 2.20 | 0.03 | 17.47 | 0.05 | -13.03 |
| Veinlet Carb. | Calcite | Apex Basalt | 3460 | A22-vein-1 | 0.12 | 0.03 | 13.50 | 0.05 | -16.88 |
| | | | | A22-vein-2 | -0.14 | 0.03 | 13.31 | 0.05 | -17.07 |
| | | | | A22-vein-3 | -0.02 | 0.03 | 13.29 | 0.05 | -17.09 |

| Sed. type | Mineral | Formation | Age | Sample | δ13C | ± | δ18O | ± | Δ |
|---|---|---|---|---|---|---|---|---|---|
| | | Mt.Ada Basalt | 3470 | MtAda-1-vein | 0.47 | 0.03 | 10.47 | 0.05 | -19.82 |
| | | Dresser Fm. | 3480 | D-2-vein-1 | -3.77 | 0.03 | 11.30 | 0.05 | -19.02 |
| | | | | D-2-vein-2 | -2.35 | 0.03 | 11.25 | 0.05 | -19.06 |
| | | | | D-2-vein-3 | -1.69 | 0.03 | 11.36 | 0.05 | -18.96 |
| Sed. Carb. | Ankerite | Euro Basalt | 3350 | E-4 | 1.88 | 0.00 | 15.24 | 0.03 | -15.19 |
| | Dolomite | Strelley Pool Fm. | 3410 | JR-Shaw-1 | 2.08 | | 15.01 | | -15.42 |
| | | | 3410 | JR-Shaw-2 | 2.52 | | 15.24 | | -15.20 |
| | | | 3410 | JR-Shaw-3 | 2.55 | | 15.31 | | -15.13 |
| | Ankerite | Dresser Fm. | 3480 | PDP | 1.26 | 0.03 | 17.83 | 0.05 | -12.69 |
| | | | | JR-TSU-1 | 2.24 | | 17.53 | | -12.98 |
| | | | | JR-TSU-2 | 2.22 | | 16.36 | | -14.11 |
| | | | | JR-TSU-3 | 2.17 | | 16.71 | | -13.77 |
| | | | | JR-TSU-4 | 1.61 | | 16.53 | | -13.95 |
| | | | | JR-TSU-5 | 2.24 | | 17.68 | | -12.83 |
| | | | | JR-TSU-6 | 2.54 | | 18.74 | | -11.80 |
| | | | | JR-TSU-7 | 2.42 | | 18.30 | | -12.23 |
| | | | | JR-TSU-8 | 2.34 | 0.05 | 18.43 | 0.07 | -12.10 |
| | | | | JR-TSU-9 | 1.21 | 0.05 | 27.10 | 0.07 | -3.69 |
| | | | | JR-TSU-10 | 1.34 | 0.05 | 15.93 | 0.07 | -14.53 |
| | | | | JR-TSU-11 | 1.61 | | 15.82 | | -14.63 |
| | | | | JR-TSU-12 | 1.61 | | 15.96 | | -14.50 |
| | | | | JR-TSU-13 | 1.49 | | 15.74 | | -14.71 |
| | | | | JR-TSU-14 | 1.38 | | 21.45 | | -9.17 |
| | | | | JR-TSU-15 | 1.10 | | 22.60 | | -8.06 |
| | | | | JR-TSU-16 | 1.78 | | 23.37 | | -7.31 |
| | | | | TSU | 1.46 | 0.03 | 15.80 | 0.05 | -14.66 |
| Sed. Carb. DB | Dolomite, calcite | | | DB | -5.10 | 0.03 | 22.79 | 0.05 | -7.88 |
| | | | | JR-Dress-1 | -5.38 | | 20.54 | | -10.05 |
| | | | | JR-Dress-2 | -6.72 | | 20.19 | | -10.40 |
| | | | | JR-Dress-3 | -6.38 | | 19.81 | | -10.77 |
| | | | | JR-Dress-4 | -6.22 | | 19.70 | | -10.87 |
| | | | | JR-Dress-5 | -6.01 | | 19.94 | | -10.64 |
| | | | | JR-Dress-6 | -4.25 | | 19.24 | | -11.32 |
| | | | | JR-Dress-7 | -5.96 | 1.72 | 1.25 | 4.76 | -28.77 |
| | | | | JR-Dress-8 | -8.07 | 0.34 | 10.50 | 1.30 | -19.79 |
| | | | | JR-Dress-9 | -3.15 | 0.07 | 19.93 | 0.15 | -10.65 |
| Stromatolite | Dolomite | Strelley Pool Fm. | 3410 | Strelley | 2.50 | 0.00 | 17.34 | 0.03 | -13.16 |
| | | | | JR-Strell-1 | 2.46 | | 13.92 | | -16.48 |
| | | | | JR-Strell-2 | 3.28 | | 15.01 | | -15.42 |

| | | | | Sample | | | | | |
|---|---|---|---|---|---|---|---|---|---|
| | | | | JR-Strell-3 | 3.38 | | 14.84 | | -15.59 |
| | | | | JR-Strell-4 | 3.32 | 0.01 | 16.64 | 0.02 | -13.84 |
| | | | | JR-Strell-5 | 2.69 | 0.01 | 15.74 | 0.03 | -14.71 |
| | | | | JR-Strell-6 | 3.30 | 0.01 | 16.56 | 0.02 | -13.92 |
| | | | | JR-Strell-7 | 3.33 | 0.01 | 16.66 | 0.02 | -13.82 |
| | | | | JR-Strell-8 | 2.58 | 0.01 | 15.91 | 0.02 | -14.55 |
| | | | | JR-Strell-9 | 3.21 | 0.01 | 16.68 | 0.03 | -13.80 |
| | | | | JR-Strell-10 | 3.38 | 0.03 | 17.47 | 0.05 | -13.04 |
| | | | | JR-Strell-11 | 3.15 | 0.01 | 16.56 | 0.02 | -13.92 |
| | | | | JR-Strell-12 | 3.19 | 0.01 | 16.63 | 0.02 | -13.84 |
| | | | | JR-Strell-13 | 3.14 | 0.02 | 16.66 | 0.03 | -13.82 |
| | | | | JR-Strell-14 | 3.03 | 0.02 | 18.34 | 0.03 | -12.19 |
| | | | | JR-Strell-15 | 3.05 | 0.01 | 16.77 | 0.02 | -13.71 |
| | | | | JR-Strell-16 | 3.26 | 0.01 | 16.84 | 0.02 | -13.65 |
| | | | | JR-Strell-17 | 3.31 | 0.01 | 16.70 | 0.03 | -13.78 |
| | | | | JR-Strell-18 | 3.04 | 0.01 | 17.17 | 0.01 | -13.33 |
| Stromatolite? | Ankerite, calcite | Isua Supracrustal Belt | 3700 | IS12-1 | 2.35 | 0.08 | 18.80 | 0.11 | -11.74 |
| | | | | IS12-2 | 1.21 | 0.08 | 19.69 | 0.11 | -10.88 |
| | | | | IS12-3 | 1.18 | 0.08 | 19.57 | 0.11 | -11.00 |
| | | | | IS12-4 | 1.11 | 0.08 | 19.59 | 0.11 | -10.98 |
| | | | | IS12-5 | 1.18 | 0.13 | 19.36 | 0.18 | -11.20 |
| | | | | IS12-6 | 1.27 | 0.08 | 19.01 | 0.11 | -11.54 |
| | | | | IS12-7 | 0.45 | 0.08 | 16.85 | 0.11 | -13.64 |
| | | | | JR-IS-1 | 0.74 | 0.08 | 19.30 | 0.11 | -11.26 |
| | | | | JR-IS-2 | 0.98 | | 19.23 | | -11.33 |
| | | | | IS-12 | 0.99 | 0.03 | 18.88 | 0.05 | -11.66 |
| | | | | IS-12-C | 1.03 | 0.03 | 19.44 | 0.03 | -11.12 |
| | | | | IS-12-Q | 0.78 | 0.03 | 19.27 | 0.03 | -11.29 |
| Metasomatic Carb. | Dolomite | | | JR-IS9 | -2.11 | | 11.46 | | -18.86 |
| | | | | IS9-1 | -2.37 | 0.08 | 11.64 | 0.11 | -18.69 |
| | | | | IS9-2 | -1.84 | 0.08 | 11.16 | 0.11 | -19.15 |
| | | | | IS9-3 | -1.74 | 0.08 | 10.93 | 0.11 | -19.38 |
| | | | | IS9-4 | -1.93 | 0.08 | 11.49 | 0.11 | -18.83 |
| | | | | IS9-5 | -2.03 | 0.00 | 11.36 | 0.03 | -18.96 |
| Rhodochrosite | Rhodochrosite | Fig Tree Fm. | 3260 | Figtree-1 | -12.74 | | 6.91 | | -23.28 |
| | | | | Figtree-2 | -10.76 | | 12.65 | | -17.71 |
| | | | | Figtree-3 | -19.34 | | -4.93 | | -34.76 |
| | | | | Figtree-4 | -12.12 | | 5.90 | | -24.26 |
| | | | | Figtree-5 | -18.23 | | -6.85 | | -36.62 |
| | | | | Figtree-6 | -23.00 | | -12.32 | | -41.93 |

| Carb. in barite | Dolomite, calcite, strontianite | Dresser Fm. | 3480 | JR-DressBart-1 | | | | | |
|---|---|---|---|---|---|---|---|---|---|
| | | | | | -18.14 | 0.20 | 9.37 | 0.50 | -20.89 |
| | | | | JR-DressBart-2 | -18.46 | 0.20 | 10.47 | 0.50 | -19.82 |
| | | | | JR-DressBart-3 | -11.37 | 0.20 | 11.32 | 0.50 | -19.00 |
| | | | | JR-DressBart-4 | -15.95 | 0.20 | 9.92 | 0.50 | -20.36 |
| | | | | JR-DressBart-5 | -11.07 | 0.20 | 10.81 | 0.50 | -19.49 |
| | | | | JR-DressBart-6 | -15.09 | 0.05 | 11.51 | 0.07 | -18.82 |
| | | | | JR-DressBart-7 | -11.81 | 0.10 | 12.06 | 0.30 | -18.28 |
| | | | | JR-DressBart-8 | -12.40 | 0.10 | 13.78 | 0.30 | -16.61 |
| | | | | JR-DressBart-9 | -9.79 | 0.10 | 12.85 | 0.30 | -17.52 |
| | | | | JR-DressBart-10 | -14.53 | 0.10 | 12.03 | 0.30 | -18.31 |
| | | | | JR-DressBart-11 | -11.20 | 0.20 | 12.30 | 0.50 | -18.05 |
| | | | | JR-DressBart-12 | -2.70 | 0.20 | 14.00 | 0.50 | -16.40 |
| | | | | JR-DressBart-13 | -10.83 | 0.10 | 12.54 | 0.30 | -17.82 |
| Carbonatite | Calcite | | | JR-C1 | -4.91 | | 7.13 | | -23.07 |
| | | | | JR-C2 | -5.84 | | 7.13 | | -23.06 |
| | | | | JR-C3 | -5.91 | | 7.00 | | -23.19 |
| | | | | JR-C4 | -3.29 | | 18.35 | | -12.18 |

**Note:**

1. $\delta^{18}O_{VPDB}=0.970017*\delta^{18}O_{VSMOW}-29.98$ (Coplen, 1988)
2. s.d. is the standard deviation calculated by multiple measurements of the in-house carbonate standard Solnhofen.
3. Abbreviations: Fm.- Formation , Mt.-Mount, Sed.-sedimentary, Carb. –carbonate;
4. The question mark in "Stromatolite (?)" indicates its controversial origin.
5. "Carb. in barite" refers to carbonate inclusions in bladed black barite.

## 4.2 Sedimentary carbonates

### 4.2.1 Laminated micritic carbonates

Laminated micritic carbonate occurs in a ca. 5 m thick sedimentary succession (Fig. 6a, b) interbedded with pillow basalts of the Dresser Formation (Fig. 2b). The micritic carbonate is predominantly brownish and finely bedded. The association with pillow basalts indicates an interval of generally quiet-water sedimentation, although the succession might preserve the oldest record of a tsunami event on Earth (Runge et al. 2022).

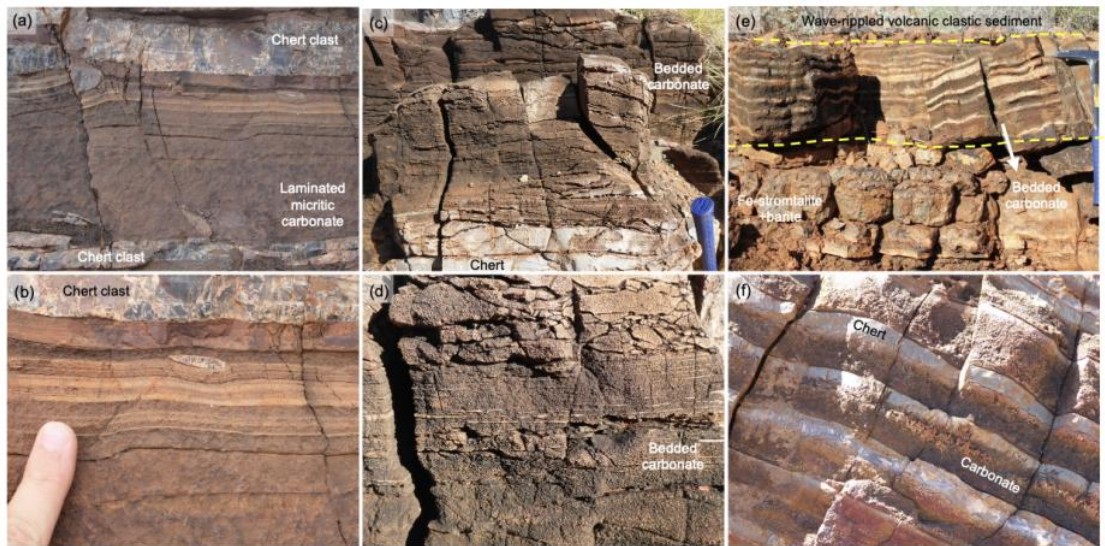

Figure 6: Outcrop photos of the bedded sedimentary carbonates from the North Pole Dome in the East Pilbara Terrane, Western Australia. (a, b) Laminated micritic sedimentary carbonate of Dresser Formation, the potential oldest reported tsunami deposit (Runge et al., 2022). (c, d) Finely bedded carbonate rock overlying the 3.35 Ga Euro Basalt. The shown bed is 10 cm thick. (e, f) Interlayered carbonate-chert beds (7 to 11 beds between the yellow dashed line) of Dresser Formation. This unit is overlain by wave rippled volcanic clastic sediment with remains of evaporitic minerals and organic films on top, while the underlying rock is bedded barite with sulfidic stromatolites atop. The length of hammer is ca. 40 cm.

The laminated micritic carbonate consists of fine-grained carbonate crystals with abundant organic clots and flakes (Fig. 7a), and locally euhedral and subhedral carbonate rhombs that have a cloudy center and clear rim enveloped by organic matter (Fig. 7b). *In situ* geochemical mappings and Raman spectra (Fig. 7c, S8; Xiang, 2023) indicate that the carbonate crystals are Mn-enriched ankerite, while that the cloudy centers consist of organic matter. Calculated Raman-based temperatures (based on Lünsdorf et al., 2017) of ~ 300–350 °C agree well with the peak metamorphic temperatures of this region (Allwood et al., 2006b; Hickman and Van Kranendonk, 2012a; Van Kranendonk et al., 2019a). The laminae are caused by changing crystal sizes and organic matter contents, with finer-grain sizes and higher organic matter contents resulting in darker colors. A similar bedded micritic carbonate is observed in samples from drilling core PDP2c (see Van Kranendonk et al., 2019b).

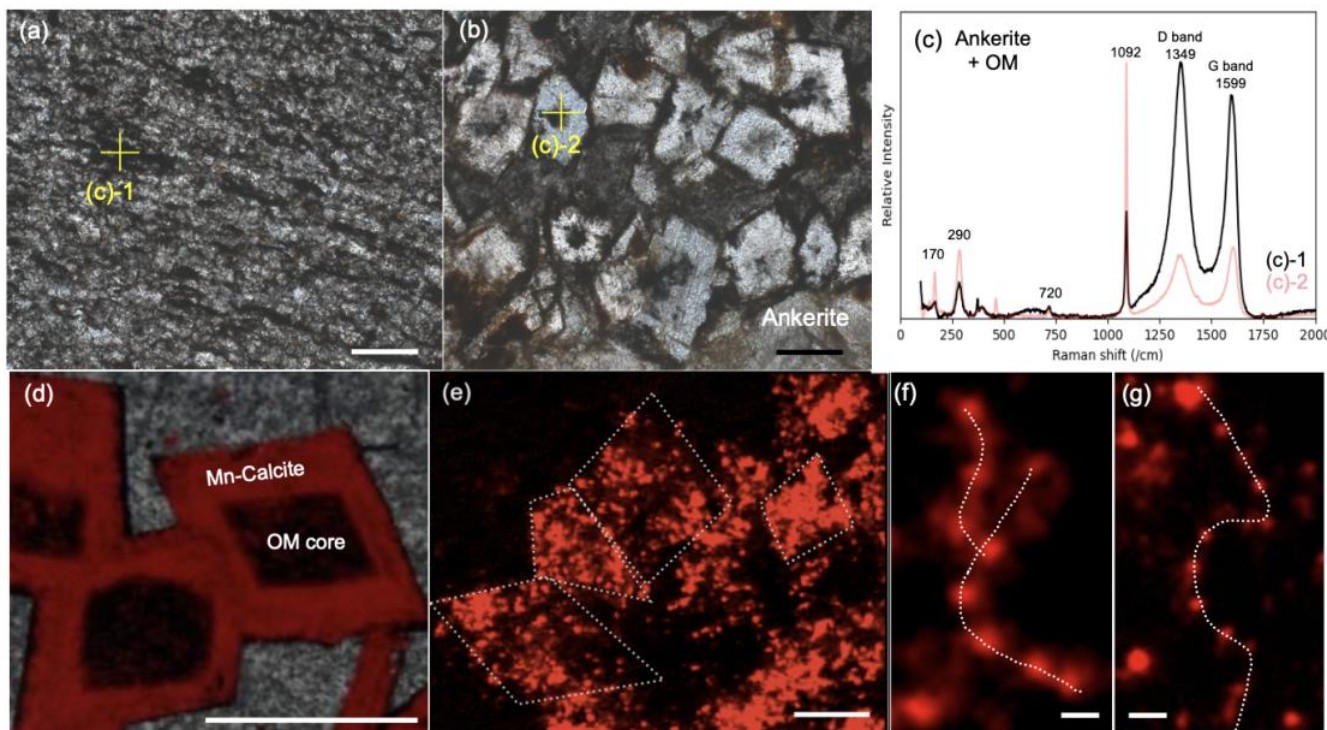

Figure 7: Spatial relationships between sedimentary carbonates and organic matter. (a) Interbed of the laminated micritic carbonate containing flakes and clots of organic material (OM). (b) Euhedral and subhedral carbonate rhombs, exhibiting organic matter in their cores and at their outer edges. (c) Raman spectra for spots in (a) and (b), supporting the presence of ankerite and organic matter. (d) Euhedral calcite rhombs with cores of organic matter are cemented by chert. (e) Close-up view of calcite rhombs showing Mn-enriched dolomite particles within the calcite crust (some crystals are indicated by dotted lines). (f, g) Arrangements of Mn-enriched dolomite particles that somewhat resemble kutnahorite formed by *Idiomarina loihiensis* strains (Rincón-Tomás et al., 2016). (a) and (b) were taken under plane-polarized light. The scale bar in (f) and (g) are 5 μm, while all others are 200 μm).

### 4.2.2 Bedded chert-carbonates

The bedded chert-carbonates are characterized by carbonate-chert couples and occur at the top of a chert layer from the Euro Basalt as well as in the Dresser Formation ("Euro bedded carbonate" and "Dresser bedded carbonate" in the following) (Fig. 6). The Dresser bedded carbonate consists of 9–11 carbonate-chert couplets with radiating crystal splays. Because of its distinct appearance, it was previously named "zebra rock" (Hickman and Van Kranendonk, 2012b; see Van Kranendonk et al. 2019b for a detailed description). Notably, it occurs between a unit of sulfidic stromatolites and bladed barite below, and wave rippled volcanoclastic sediments above (Fig. 6e).

Individual carbonate-chert couplets consist of fining-upward successions of euhedral to subhedral carbonate rhombs in a chert matrix (Fig. 8a, b). In the Dresser bedded carbonate, carbonate rhombs are commonly dolomite and calcite (Fig. S4c), and

clusters of radiating carbonate crystal splays occur at the base of each couplet (Figs. 6f, 8a). Some carbonate rhombs have an organic core (Fig. 7d) and show a strong patchy Mn enrichment pattern under CL (Fig. 7e–g), somewhat similar to kutnahorite [Ca(Mn,Mg,Fe)(CO₃)₂] formed by modern *Idiomarina loihiensis* (ɣ-proteobacteria) (Rincón-Tomás et al., 2016). The euhedral to subhedral carbonate rhombs and the highly porous chert matrix (Fig. 8b) indicate low compaction after deposition. In the Euro bedded carbonate, organic matter is rare and only interbedded between carbonate crystals (ankerite; Fig. S4d, e). Although the Euro bedded carbonate exhibits the repeated grading of ankerite rhombs in a chert matrix, pressure dissolution features associated with ankerite crystals and the nonporous microcrystalline chert matrix imply a stronger post-depositional compaction (Fig. S9a, b).

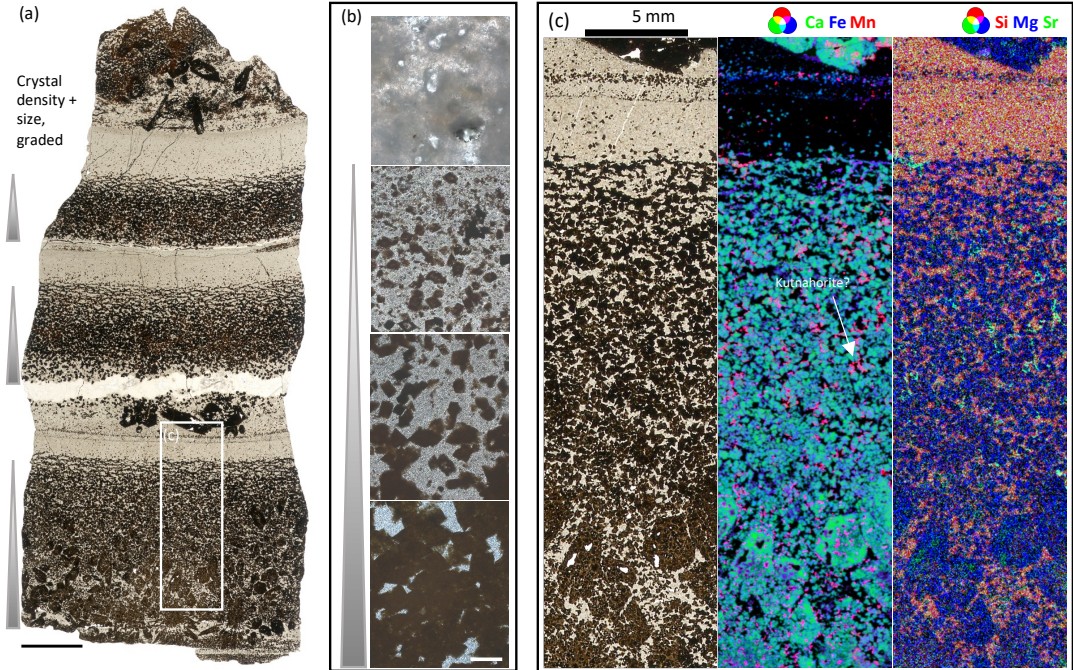

**Figure 8: The bedded sedimentary carbonate-chert rock from Dresser Formation. (a) The scan image (transmitted light) of thin section shows repeated graded carbonate layers with the crystal size and density decreasing upwards. One layer is shown discontinuously in (b), that euhedral carbonates grade into the chert layer of high-porosity. The rectangle area is magnified in (c). (c) The false-colour overlapping images show that the Mn-dolomite particles distribute on the edge of calcite crystals. Scale bar in (a) is 10 mm and in (b) is 200 μm. The images of each element are shown in Fig. S10.**

*In situ* geochemical mappings reveal that the Dresser bedded carbonate predominantly consists of Fe-enriched dolomite and calcite with Mn-enriched dolomite particles along its edges (Fig. 8c), in agreement with the observed CL patterns. The Euro Basalt related bedded carbonate comprises Mn-enriched ankerite (Fig. S9).

### 4.2.3 δ¹³C and δ¹⁸O values of sedimentary carbonates

Laminated micritic carbonate and the Euro Basalt related bedded carbonate show $\delta^{13}C$ values between 1.10 and 2.55 ‰ (mean = 1.85 ± 0.48‰) and $\delta^{18}O$ values between -15.42 and -3.69 ‰ (mean = -12.75 ± 3.00 ‰) (Table 1). The Dresser bedded carbonate, in contrast, exhibits the more negative $\delta^{13}C$ values ranging from -8.07 to -3.15 ‰ (mean = -5.72 ± 1.36 ‰) and $\delta^{18}O$ values ranging from -28.77 to -7.88 ‰ (mean = -13.11 ± 6.32 ‰) (Table 1).

### 4.3 Stromatolites

The stromatolitic carbonates were located from the second member of the Strelley Pool Formation. Stromatolite morphologies and arguments for biogenicity have been reported in detail elsewhere (Allwood et al., 2006a, 2007; Van Kranendonk et al. 2003; Van Kranendonk, 2011; Duda et al., 2016; Viehmann et al., 2020). Briefly, stromatolites show a high morphological diversity, ranging from coniform and finely laminated to large domical forms, and overly centimeter-sized carbonate fans (Fig. 9).

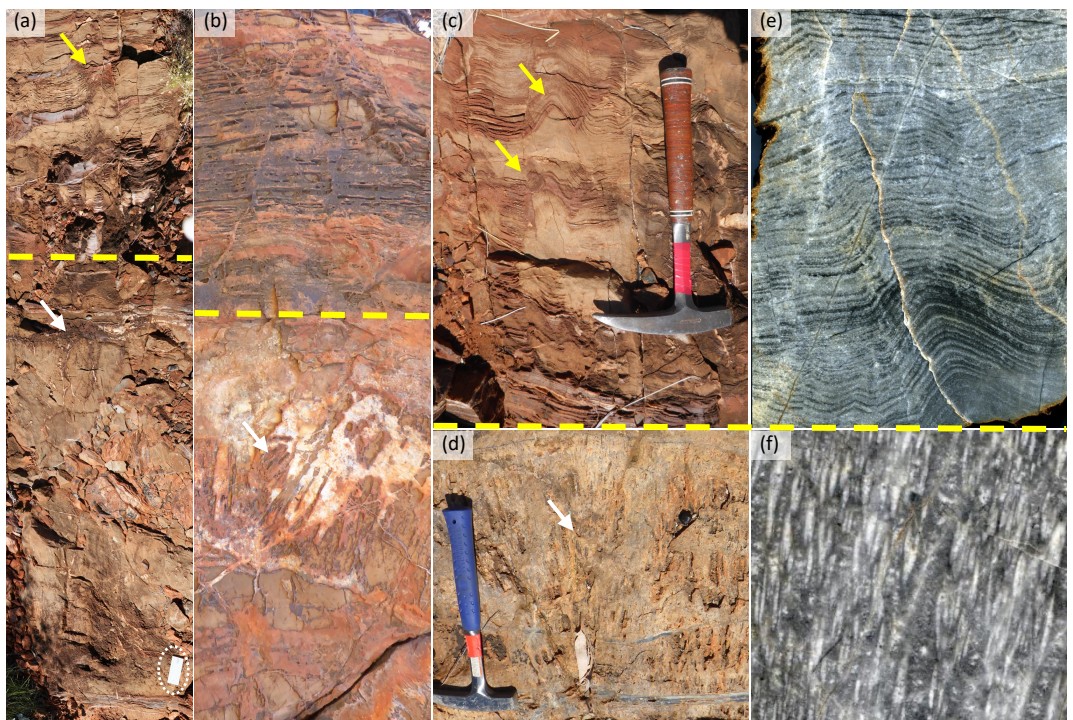

**Figure 9: Photos of stromatolites from the Strelley Pool Formation near the Trendall site in the East Pilbara Terrane, Western Australia. The yellow dashed line marks the boundary between the upper unit with conical stromatolites (yellow arrow) and the lower unit with carbonate fans (white arrow). (a) Composite photo of the outcrop. The ruler (white dotted circle) is 15 cm in length. (b) Close-up view of an outcrop showing the layered stromatolite consisting of carbonate (weathered and partly absent) and chert (dark beds), atop the large carbonate fan on a chert matrix. (c, d) Close-up view of the conical stromatolites (c) and carbonate fans**

(d). (e, f) Cross-section views of samples corresponding to (c) and (d), respectively. The length of the brown and blue hammers in (c) and (d) are ~30 cm and ~40 cm, respectively.

The studied sample is a silicified coniform stromatolite with alternating laminae of equigranular anhedral dolomite that often preserves organic matter. The laminae margin contains euhedral dolomite overgrowth (Fig. S4f). Detailed cement stratigraphy involving CL microscopy indicate the presence of at least three dolomite generations (Fig. S11), in line with previous works (Allwood et al., 2009, 2010; Flannery et al., 2018). The stromatolites occur atop large, chert cemented carbonate fans (~ 40 cm) situated on a chert layer (Fig. 9). The carbonate fans encompass fusiform dolomite aggregations (Fig. 9f).

The stromatolites show $\delta^{13}C$ values ranging from 2.46 to 3.38 ‰ (mean = 3.08 ± 0.30 ‰) and $\delta^{18}O$ values ranging from -16.48 to -12.19 ‰ (mean = -14.03 ± 0.98 ‰), consistent with data reported in Lindsay et al. (2005) and Flannery et al. (2018).

## 5 Discussion

### 5.1 Formation pathways of the EPT carbonates

The tectonic model of the EPT involves a volcanic plateau characterized by surface topographical changes, as indicated by pillow basalt successions and shallow water deposits (Smithies et al., 2003, 2005, 2007a, b; Van Kranendonk, 2006; Van Kranendonk et al, 2007a, b, 2019a). Our survey demonstrates that carbonates occur in very different EPT environments, ranging from deep marine settings to terrestrial ponds, which all have been differently influenced by hydrothermal processes.

#### 5.1.1 Carbonate abiotically precipitated from hydrothermal fluids

Carbonates associated with Archean pillow basalts are well known from various localities worldwide (Roberts, 1987; Veizer, 1989a, b; Kitajima et al. 2001; Nakamura and Kato, 2004). Today, the formation of such carbonates is triggered by fluctuations in alkalinity, salinity and water temperature (Degens et al., 1984; Kempe, 1990; Reitner et al. 1995b; Flügel, 2010), i.e., the underlying processes are controlled by abiotic parameters. In modern settings, carbonates usually precipitate from low- to moderate-temperature hydrothermal fluids during the latest stage of seafloor alteration (Bach et al., 2001, 2003, 2011; Coogan and Gillis, 2013); as a consequence, they tend to be more abundant in older crusts (Gillis et al., 2001; Heft et al., 2008; Coogan and Gillis, 2013). The precipitation of Ca-Mg-Fe carbonates and formation of silica-bearing fluids linked to basalt-fluid interactions has also been demonstrated by experimental work and numerical simulations (~22 to 350 °C) (Gysi and Stefánsson, 2011; Gudbrandsson et al., 2011; Stockmann et al., 2011; Galeczka et al., 2013a, b, 2014; McGrail et al., 2017; Menefee et al., 2018; Wolff-Boenisch and Galeczka, 2018; Xiong et al., 2018; Voigt et al., 2018).

There is no evidence for a potential biological influence on the formation of EPT basalt-associated carbonates as for instance organic remains. At the same time, precipitation of these carbonates could have been abiotically triggered by infiltration of $CO_2$-enriched seawater and/or basalt-water interactions under hydrothermal conditions, which result in a higher alkalinity and higher cation concentrations (Fig. 10). Indeed, fracture-filling calcite shows the lowest $^{87}Sr/^{86}Sr$ ratio (0.700596) and REE+Y pattern that is considered typical of Archean seawater (Appendix B; for further details see Xiang, 2023), indicating the

percolation of seawater-derived $CO_2$-rich fluids through basaltic crust 3.5 Ga ago (Kitajima et al. 2001; Nakamura and Kato, 2002; Yamamoto et al., 2004). In this light, $\delta^{13}C$ signatures of fracture-filling calcites ($2.18 \pm 0.13$ ‰ on average; Fig. 11) may reflect Archean seawater, while $\delta^{13}C$ signatures of interstitial carbonates ($0.22 \pm 0.98$ ‰ on average; Fig. 11) indicate admixture of hydrothermally derived mantle-derived carbon ($\delta^{13}C$ of -5 to -6 ‰; Degens et al. 1984, Hayes and Waldbauer 2006). This is reflected in the common lower $\delta^{13}C$ values of veinlet carbonates than the interstitial carbonates (Fig. S3).

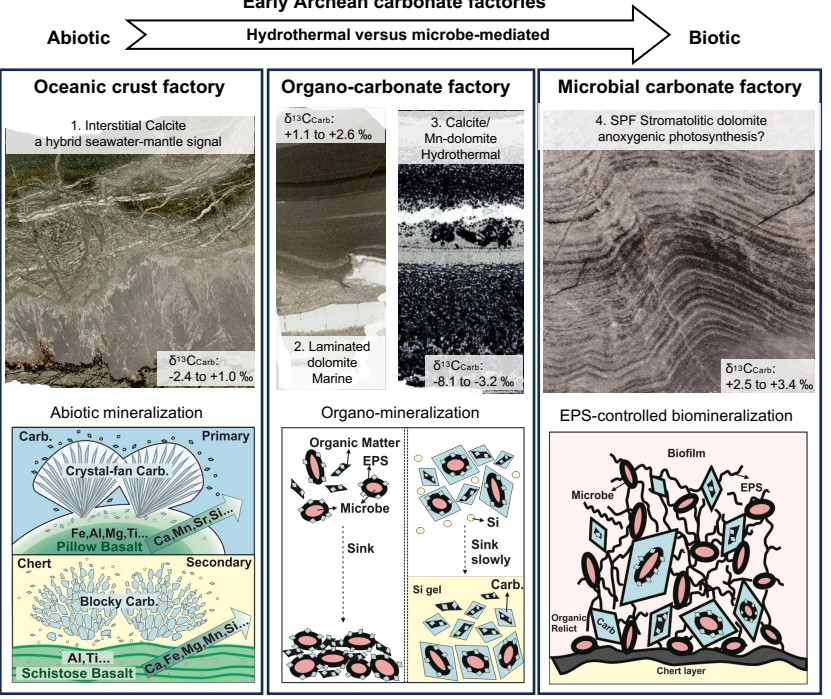

**Figure 10: The lithological features and formation pathways of the three carbonate factories in the early Archean, including an oceanic crust factory, an organo-carbonate factory and a microbial carbonate factory. Carbon precipitation in the oceanic crust factory is an abiotic and inorganic process driven by seawater-basalt interaction, which produce hydrothermal fluids with high carbonate alkalinity and high cation concentrations. Carbonate precipitation in the organo-carbonate factory is linked to organic macromolecules (i.e., organo-mineralization). Carbonate precipitation in the microbial carbonate factory occurs through EPS-controlled biomineralization, with anoxygenic photoautotrophs being a likely source of the EPS (adapted from Reitner et al., 2001). (Abbreviations in figure: "Carb.", "Carb"- carbonate; "EPS"- extracellular polymeric substances)**

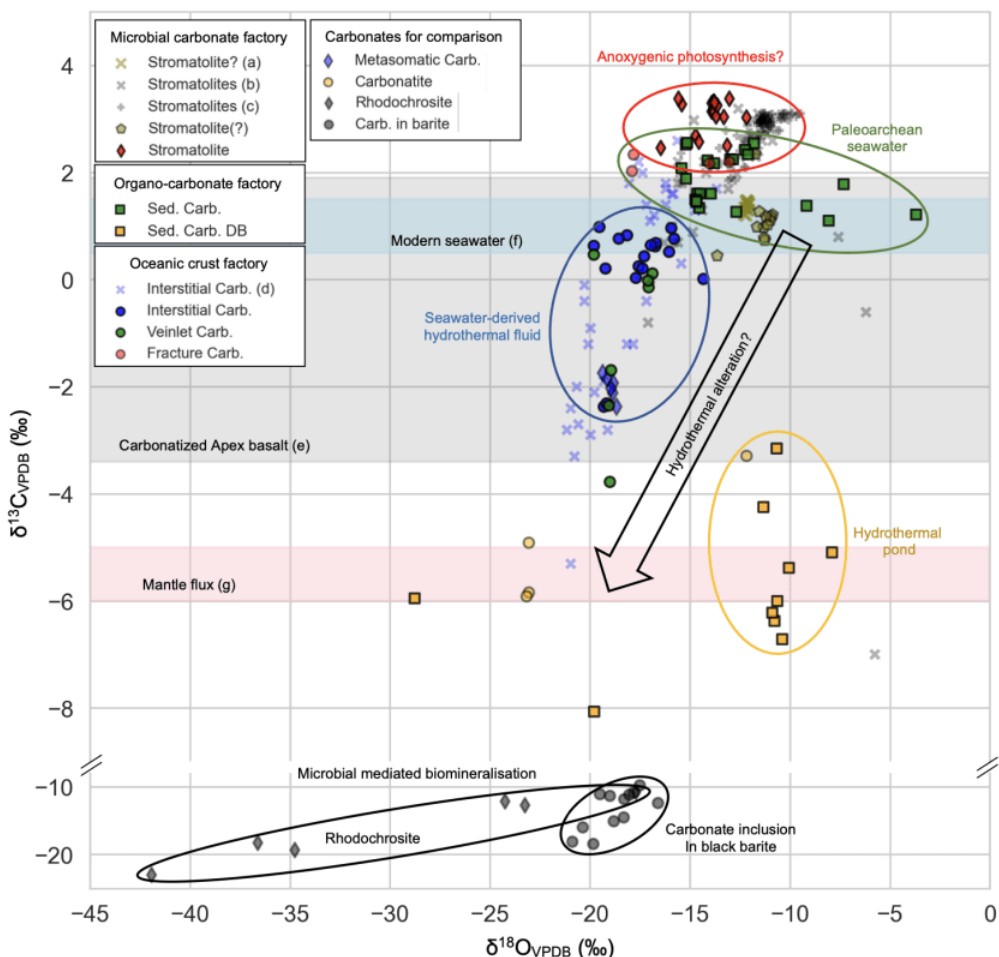

**Figure 11: Stable carbon and oxygen isotopic (δ¹³C, δ¹⁸O) compositions of early Archean carbonates (Carb.). In tendency, δ¹³C and δ¹⁸O values decrease from the stromatolite through marine sedimentary carbonate (Sed. Carb.) to interstitial carbonate, possibly reflecting increasing admixture of mantle-derived carbon. Own data (square, circle, diamond, and pentagon symbols) are given in Table 1. Reference data (including cross and plus symbols) from (a) Nutman et al. (2016), (b) Lindsay et al. (2005), (c) Flannery et al. (2018), (d) Shibuya et al. (2012), (e) Nakamura and Kato (2004), (f) Kroopnick (1980) and Tan (1988), (g) Degens et al. (1984), and Hayes and Waldbauer (2006). Note that "Carb. in barite" indicates carbonate inclusions in black barites from the ~3.5 Ga Dresser Formation (Western Australia), and that question marks in sample labels highlight the controversial biogenicity of the material.**

The mixture of different fluids is also supported by $^{87}Sr/^{86}Sr$ ratios of primary interstitial calcite associated with the Apex basalt (0.703094 ± 0.000979), laying between those of early Archean seawater and Apex pillow basalt (0.700596 and 0.706337 ± 0.000954, respectively; Appendix B; for further details see Xiang, 2023). On the other hand, the precipitation of calcite in vesicles and veins of basalts, as well as the formation of acicular calcite crystal-fans growing at pillow margins (Fig. 3a), could

have been driven by an elevated alkalinity and higher Ca levels, which derived from hydrothermal basalt-water interactions. The observed blocky and massive interstitial calcite or ankerite (Figs. 4, 5) probably resulted from recrystallization and ankeritization of primary calcite precipitates, driven by Mg-, Fe-, Al-, Si-, and Mn-enriched fluids deriving from hydrothermal chlorite breakdown in the basalts (Fig. 3). In summary, carbonates associated with pillowed basalts are inferred to have precipitated abiotically on and below the seafloor (Fig. 12).

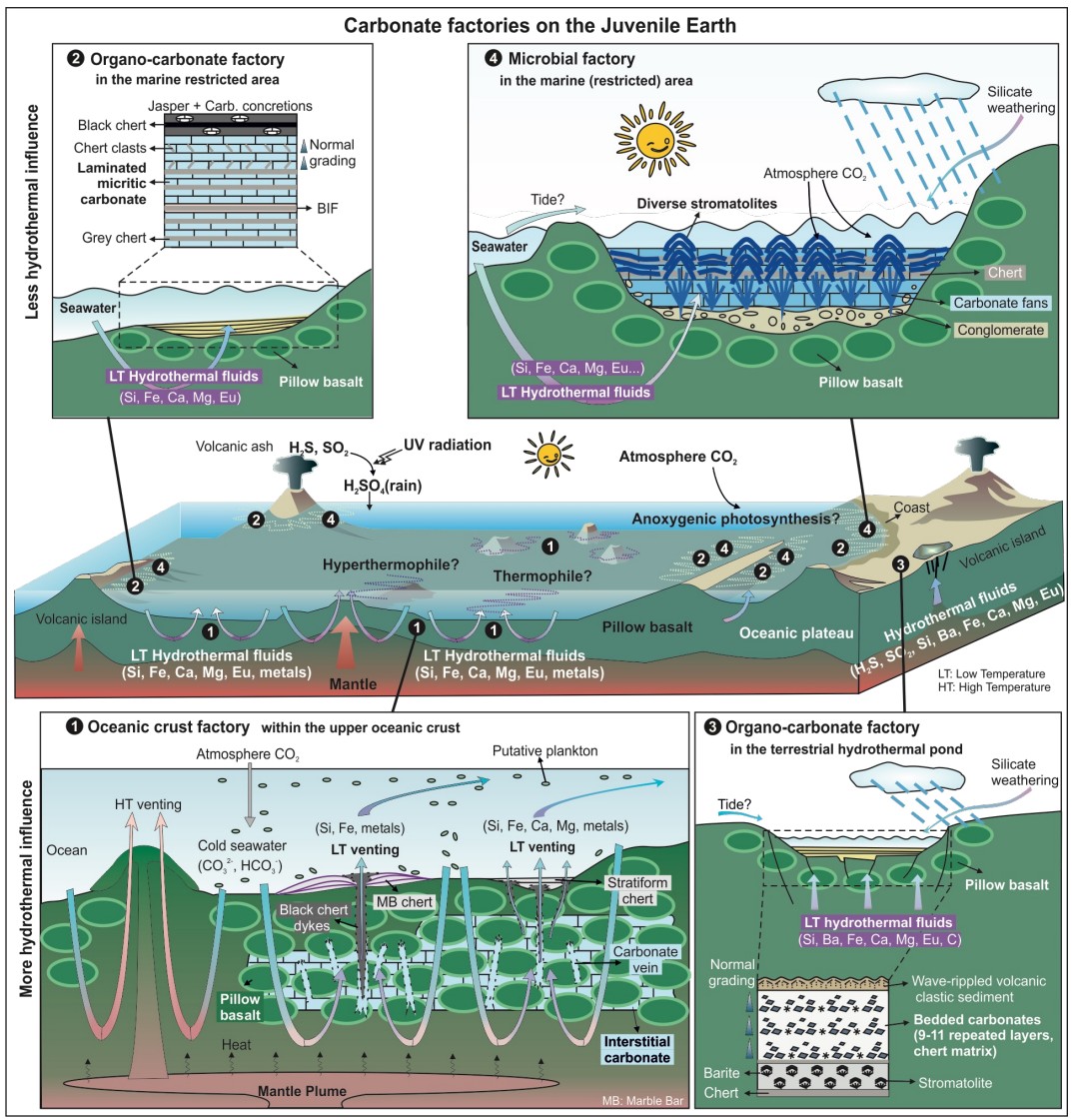

**Figure 12: The possible localities of the three early Archean carbonate factories (adapted from Nisbet and Sleep, 2001; Runge et al., 2022). The oceanic crust factory commonly occurs in deeper marine environments within the upper oceanic crust (number 1). The organo-carbonate factory may operate in diverse environments where organic matter (biotic and/or abiotic) is abundant and fluids are supersaturated in $Ca^{2+}$ and $CO_2$, and which are intermittently influenced by hydrothermal fluids (number 2 and 3). The**

**microbial factory likely forms in photic, relatively restricted, shallow marine environments like lagoons on the slope or platform, with minor detritus and rare hydrothermal inputs (number 4).**

### 5.1.2 Bedded sedimentary carbonates – a product of organo-mineralization

EPT bedded sedimentary carbonate rocks preserve abundant organic remains, for instance occurring as dispersed flakes and clots within micrite (Fig. 7), perhaps indicating a genetic relationship. Organic matrices and compounds inherited from living organisms may retain mineralizing properties (Défarge and Trichet, 1995; Trichet and Défarge, 1995). To distinguish minerals formed through mineralization linked to organic matrices and compounds from those whose formation is induced by living organisms, the terms "organomineral" and "organo-mineralization" were introduced at the 7th International Symposium on

Biomineralization in 1995 and further developed in the following decade (Défarge and Trichet, 1995; Reitner et al., 1995b, 1997; Arp et al., 1999, 2001, 2003; Neuweiler et al., 1999;Riding, 2000; Pratt, 2001; Reitner, 2004; Gautret and Trichet, 2005), before being finally confirmed in following studies (Perry et al., 2007, 2009; Défarge et al., 2009; Altermann et al., 2009).

Fine-grained carbonates such as micritic calcite or dolomite are typical products of organo-mineralization. Indeed, micrite can

form autochthonously (i.e., "automicrite"), involving Ca-binding by aspartic acid- and glutamic acid-rich (briefly, Asp- and Glu-) proteinous macromolecules and negatively charged polysaccharids initiating carbonate crystal nucleation (Reitner et al. 1995a, b, c; Reitner and Neuweiler, 1995; Trichet and Défarge, 1995). It is consequently termed "organomicrite", which is a distinct subset of automicrite (Reitner et al., 1995b). Organomicrites are widespread in Phanerozoic carbonate depositional systems, particularly important in microbial mats and biofilms (Reitner et al., 1995a, b, c; Riding, 2000; Schlager, 2000 2003;

Reitner and Thiel, 2011; Reijmer, 2021), as well as in the SPF stromatolites. However, it is important to highlight that organo-mineralization is not restricted to organic matter from biological sources (Défarge et al., 2009); in fact, laboratory experiments indicated that abiotic organic matter as e.g. from the Murchison CM2 meteorite can also mediate carbonate precipitation (Reitner, 2004).

Organomicrites are abundant in some EPT facies. The precipitation and sedimentation of organomicrites result in the finely

laminated carbonate deposits (Figs. 7a, 10). However, carbonate precipitation was more complicated in case of fine-graded, bedded chert-carbonates (Figs. 7b-d, 10). In this case, organomicrites can serve as new nucleation centers for the development of euhedral carbonate rhombs. Under low-energy conditions, the fine-grained carbonate crystals precipitate slowly, displaying pronounced normal grading (Figs. 8b, 10). On the other hand, occurrence of Fe/Mn-enriched carbonates and the chert matrix is indicative of involvement of Si-bearing hydrothermal fluids in this process (Fig. 8). Hydrothermal fluids provided significant

alkaline metals and silicon. Silicon precipitated as opal-A, a soluble hydrated amorphous silica phase that deposited as siliceous gel, which was converted to chert during diagenesis (Ledevin, 2019). The cyclical repeat of this process resulted in the formation of bedded sedimentary carbonate (Fig. 8a).

Some EPT bedded sedimentary carbonates (except the Dresser bedded carbonates) show an average $\delta^{13}C$ value of 1.85 ± 0.48‰ (Fig. 11). These is consistent with other reports on Strelley Pool stromatolites (Lindsay et al., 2005; Flannery et al.,

2018) and within the range of modern seawater (Kroopnick, 1980; Tan, 1988), reflecting their formation in marine environments. At the same time, $\delta^{13}$C values of the Dresser bedded carbonates are relatively depleted (-5.72 ±1.36 ‰ on average), in good accordance with $\delta^{13}$C signatures of carbonatites (-4.99 ± 1.22 ‰ on average) (Fig.11), indicating hydrothermal admixture of mantle-derived carbon. The occurrence of rippled volcanic clastic sediments atop (Fig. 6e) indicates a shallow water environment. The clusters of radiating calcite crystals at the base of each carbonate-chert layer (Figs. 6f, 8a), which were initially proposed to be gypsum or aragonite (Runnegar et al., 2001; Van Kranendonk et al., 2008; Otálora et al., 2018), are likely indicative of evaporitic conditions. The $\delta^{13}$C values and field relationships imply that the Dresser bedded carbonates perhaps formed in a terrestrial hydrothermal pond with intermittent inputs, akin to recently identified hot spring deposits (Djokic et al., 2017, 2021). Hence, bedded sedimentary carbonates formed across a spectrum of environments, ranging from shallow marine to terrestrial settings.

### 5.1.3 Stromatolites formed through microbial activity

Stromatolites, first described by Kalkowsky (1908), are defined as laminated benthic microbial deposits (Hofmann, 1973; Buick et al., 1981; Riding, 1999; Flügel, 2010). Although the biogenicity of early Archean stromatolites is commonly controversial, that of stromatolites from the Dresser Formation and the SPF has been widely accepted (Lambert et al. 1978; Van Kranendonk 2006, 2007; Allwood et al. 2006a, 2007, 2009; Marshall et al. 2007; Wacey, 2010; Bontognali et al., 2012; Duda et al. 2016; Flannery et al. 2018; Mißbach et al., 2021; Weimann et al., 2024). Carbonates associated with SPF stromatolites are thought to be related to microbial processes (Van Kranendonk 2007, 2011; Lepot, 2020).

Stromatolites typically form through biologically induced or controlled mineralization within microbial mats or biofilms, commonly related to physicochemical gradients and/or organic substances providing nucleation sites for mineral precipitation (Reitner et al., 2000). In any case, extracellular polymeric substances (EPS) secreted by microorganisms play a key-role in mineralization (Decho, 2011; see in Fig. 10). Certain functional groups of organic substances in the EPS (e.g. Asp- and Glu-rich macromolecules) efficiently bind and sequester divalent cations such as $Ca^{2+}$ and $Mg^{2+}$, thereby inhibiting their combination with carbonate anions and subsequent precipitation (Reitner et al. 1995a, b, c). This process is somewhat similar to organo-mineralization, which involves a mineralization of organic matrices and compounds decoupled from the source organisms or of abiotic origin (Trichet and Défarge, 1995; Défarge et al., 2009; Défarge, 2011). In case of EPS-controlled biomineralization, carbonate nucleation and growth are extracellular processes, which are triggered by the metabolic activity of microorganisms and related changes in the immediate environment (Heim, 2011).

EPS-controlled biomineralization might have played a role in case of the SPF stromatolites, as supported by $\delta^{13}$C signatures of carbonates. More specifically, $\delta^{13}$C values of carbonates from SPF stromatolites (3.08 ± 0.30 ‰ on average) are higher than those of the interstitial carbonates (0.22 ± 0.98 ‰ on average) and the sedimentary carbonates (1.85 ± 0.48‰ on average). This difference is well in line with a sequestration of $^{12}$C by photoautotrophic microorganisms in the microbial mats, resulting in an enrichment of $^{13}$C in the environment and, consequently, in the carbonate (e.g., Arp et al., 2011). Additionally, the positive $\delta^{13}$C values of the SPF stromatolites are distinctive to those of rhodochrosite from the Fig Tree Group and of carbonate

inclusions in black barites of the Dresser Formation (-16.03 ± 4.86 ‰ and -12.56 ± 4.10 ‰ on average, respectively; Fig. 11), which are assumed to precipitate from microbial biomineralization and hydrothermal carbon. Flannery et al. (2018) reported

a substantial $\delta^{13}C_{org}$ fractionation in SPF stromatolites and fan-like carbonates (similar to the materials investigated herein), ranging from -29 to -45 ‰. It is documented to be compelling evidence for the coexistence of autotrophic possibly anoxygenic photosynthesis or predominantly heterotrophic metabolisms alongside the Calvin-Benson-Bassham (CBB) cycle (Flannery et al., 2018). Anoxygenic phototrophs appear to be plausible candidate microorganisms, given that they likely appeared about 3.8–3.4 billion years ago (Awramik, 1992; Brasier et al., 2006; Moore et al., 2017; Lepot, 2020). Taken together, carbonates

associated with SPF stromatolites precipitated in shallow marine environments, perhaps lagoon-like, relatively restricted basin (see Fig.12).

## 5.2 Early Archean carbonate factories – implications

Depending on the formation mechanisms, the identified EPT carbonates can be assigned to three carbonate factories: (i) an oceanic crust factory, (ii) an organo-carbonate factory, and (iii) a microbial carbonate factory. The formation pathways and

500 depositional environments are summarized in Table 2. The oceanic crust factory includes abiotically formed carbonates such as Mn- or Sr-enriched calcite and ankerite that are associated with pillow basalts within the upper oceanic crust. Carbonates in this carbonate factory precipitated from $CO_2$-rich seawater-derived hydrothermal fluids characterized by a high alkalinity and high cation loads. The organo-carbonate factory is dominated by authigenic carbonates formed through taphonomy-controlled organo-mineralization (i.e. organomicrites). Importantly, and in contrast to the microbial carbonate factory, the

505 involved organic matter can be of either biological or abiotic origin. For this reason, precipitates assigned to this carbonate factory formed in various environments, ranging from shallow marine to terrestrial settings. The microbial carbonate factory is somewhat similar to the organo-carbonate factory, but specifically refers to EPS-controlled carbonate precipitation, that is, mineralization of biologically derived organic substances. However, as in case of the organo-carbonate factory, organomicrite is formed as a typical product. Since this carbonate factory is directly linked to biological activity, the assigned precipitates

typically occur in the photic, relatively restricted, shallow marine environments like lagoons. Given that most of these carbonates formed in shallow-water environments under anoxic conditions, anoxygenic phototrophs appear a plausible source of biological organic matter, but this remains to be tested in future studies.

**Table 2: Features of the three carbonate factories in the early Archean**

| Features | Oceanic crust factory | Organo-carbonate factory | Microbial carbonate factory |
|---|---|---|---|
| Primary lithology | Acicular crystal-fan calcite | Organomicrite, calcite or ankerite crystals of various size, on a chert matrix | Laminated dolomite layers cemented by chert |

| | | | |
|---|---|---|---|
| Secondary lithology | Sparite, blocky, massive calcite and ankerite | Anhedral dolomite crystals showing compaction and pressure dissolution | Several generations of dolomites, including prismatic dolomite cement |
| Organic Materials (OM) | Absent | Abundant | Abundant |
| Origins of OM | - | Abiotic to biogenic | Biogenic |
| Hydrothermal inputs | Dominant | Common | Rare |
| Main origins of carbonate | Inorganic precipitation from seawater or seawater-derived hydrothermal fluids | Taphonomy-controlled organo-mineralization | EPS-controlled microbial mineralization |
| Evaporite minerals | Absent | Common to rare | Common to rare |
| Silicon in fluid | Source/sink | Sink | Sink |
| Siliciclastic sediments | Absent | Common | Common |
| Depositional setting | Deeper marine within the upper ocean basaltic crust | Diverse, shallow ocean to terrestrial hydrothermal pond | Photic shallow marine slope/ platform |

In case of all three carbonate factories, hydrothermal fluids play a key role in the formation and preservation of carbonate precipitates. The precipitation of carbonates might for instance be directly driven by basalt-alteration, or rather indirectly by providing a nutrient source for EPS-forming microorganisms. Preservation of carbonates is commonly promoted by hydrothermally driven silicification in the environment or during early diagenesis, which is well known for carbonaceous materials in early Archean rocks (Glikson et al., 2008; Alleon et al., 2016; Duda et al., 2016, 2018; van Zuilen, 2019; Hickman-Lewis, 2019; Ledevin, 2019; Lepot, 2020). Our study shows that such processes are also critical for the preservation of very delicate features in carbonates, allowing for the identification of precipitates formed in the three carbonate factories.

**5.3 Carbon sinks during the early Archean**

The Earth's surface carbon cycle is largely determined by mantle-derived atmospheric $CO_2$ as the single carbon source and biologically-derived organic carbon burial and carbonate sedimentation as the two main carbon sinks (Gislason and Oelkers, 2014; Hoefs, 2018; Shields, 2019). The carbon isotope ratios of the source ($\delta^{13}C$ of mantle $CO_2$ = -5 ‰) and the carbon sinks ($\delta^{13}C$ of organic matter = -25 ‰, $\delta^{13}C$ of seawater carbonate = 0 ‰) have remained largely constant throughout Earth history

(Schidlowski, 1988). Following a simple carbon isotope mass balance equation (see Eq. 2), in which the carbon input to the surface reservoirs is balanced by these two sinks only, it follows that the fraction of organic carbon that is buried relative to the total carbon input ($f_{org}$) has remained constant since the early Archean, with a value of ca. 0.2 (Hayes et al., 1999). Since the burial of organic carbon is directly linked to the flux of oxygen to the atmosphere (Campbell and Allen, 2008; Hayes and Waldbauer, 2006), a constant $f_{org}$ would imply that the carbon cycle alone cannot explain the rise of atmospheric oxygen at 2.4 Ga (Krissansen-Totton et al., 2015). Rather, processes affecting the oxygen sink would have to be invoked, such as e.g. a change from predominantly subaerial volcanism to submarine volcanism (Kump and Barley, 2007), mantle redox evolution (Kasting et al., 1993), or atmospheric hydrogen escape (Catling et al., 2001; Claire et al., 2006; Zahnle et al., 2013). Alternatively, the classical carbon isotope mass balance equation requires refinement, since it does not take into account key carbonate sinks such as oceanic crust carbonatization (briefly OCC, Bjerrum and Canfield, 2004) and authigenic carbonate that precipitates inorganically in situ (Schrag et al., 2013).

In the early Archean, when continental crustal mass was much less than today (Taylor and McLennan, 1981; Arndt, 1999; Flament et al., 2008; Cawood et al., 2013; Korenaga, 2021) and thus terrestrial silicate weathering was minimal or even absent, OCC would have been the dominant carbonate sink. The fraction of OCC of the total inorganic carbon sink ($\lambda$), was estimated by Bjerrum and Canfield (2004) to have evolved from 0.95 in the early Archean to 0.39 at the onset of the Phanerozoic. Taking this into account in an extended carbon isotope mass balance equation (Bjerrum and Canfield, 2004), it is shown that the fraction of organic carbon burial $f_{org}$ was likely much lower than today, with values closer to 0.1. Critically important in this extended mass balance is the carbon isotope ratio of the OCC, which was estimated by Bjerrum and Canfield (2004) based on micritic siderites in deep-water BIF's ($\delta^{13}C$ of -5 to -3 ‰) and modern seafloor hydrothermal carbonate deposits ($\delta^{13}C$ of -2 to 0 ‰).

The carbon isotope ratios that we report here for the three carbonate factories can now be used to verify this carbon isotope mass balance calculation. Bjerrum and Canfield (2004) postulated that total inorganic carbon removal comprises two primary fluxes, sedimentary carbonate carbon (SCC) and OCC (Fig. 13). The OCC-mediated carbon removal is expressed as a fraction ($\lambda$) of the total. Based on this, isotopic mass balance equations may be written as:

$$\delta^{13}C_{in} = \delta^{13}C_{org}f_{org} + \delta^{13}C_{IC}(1 - f_{org}) \qquad \text{(Eq.2)}$$

$$\delta^{13}C_{IC} = \delta^{13}C_{OCC}\lambda + \delta^{13}C_{SCC}(1 - \lambda) \qquad \text{(Eq.3)}$$

where $\delta^{13}C_{in}$ represents the isotopic ratio of carbon entering oceans; $\delta^{13}C_{IC}$, $\delta^{13}C_{OCC}$ and $\delta^{13}C_{SCC}$ respectively signify the isotopic ratios of total inorganic carbon precipitated as carbonate, carbonate in ocean crust and sediment; $f_{org}$ denotes the fraction of organic carbon ($\delta^{13}C_{org}$) removed from total oceanic carbon. Then Eq. 2 can be rewritten in conjunction with Eq. 3 as:

$$\delta^{13}C_{in} = \delta^{13}C_{SCC} + f_{org}\left(\delta^{13}C_{org} - \delta^{13}C_{SCC}\right) + \lambda(1 - f_{org})(\delta^{13}C_{OCC} - \delta^{13}C_{SCC}) \qquad \text{(Eq.4)}$$

Given $\Delta_S = \delta^{13}C_{OCC} - \delta^{13}C_{SCC}$ and $\Delta_b = \delta^{13}C_{org} - \delta^{13}C_{SCC}$, then the following isotopic mass balance is obtained:

$$\delta^{13}C_{in} = \delta^{13}C_{SCC} + f_{org}\Delta_b - \lambda(1 - f_{org})\Delta_S \qquad \text{(Eq.5)}$$

where $\Delta_S$ and $\Delta_b$ represent the isotopic differences between the inorganic carbon removed by OCC and SCC, organic carbon and SCC, respectively.

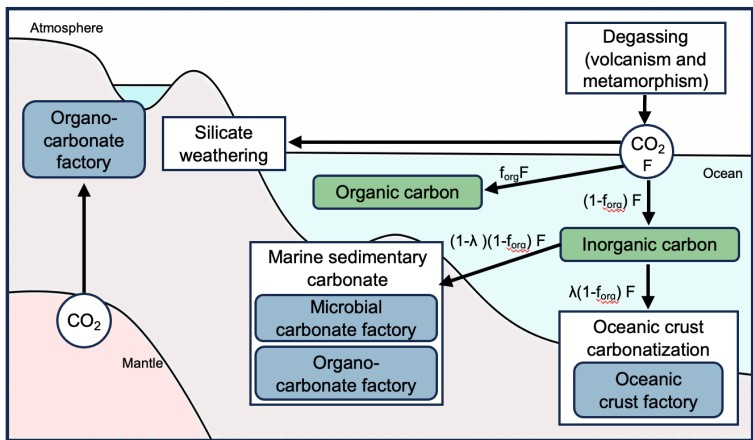

**Figure 13: Simplified carbon sinks in the early Archean. Carbon entering the ocean (F) was partitioned: a fraction ($f_{org}$) converted**
**to organic carbon, while the remainder ($1-f_{org}$) became inorganic carbon as carbonates. Carbonate carbon was further divided, with a fraction ($\lambda$) sequestered in the oceanic crust factory and the rest processed in microbial and organo-carbonate factories. The green boxes represent two carbon sinks in ocean; blue boxes depict three carbonate factories; solid arrows indicate carbon flux (adapted from Mills et al., 2014).**

By utilizing the $\delta^{13}C$ range of carbonate in the oceanic crust factory (-2.37 to 0.99 ‰) as a reference for OCC, and those of
570 marine sedimentary carbonate from the organo-carbonate (1.10 to 2.55 ‰) and microbial carbonate factories (2.46 to 3.38 ‰) combined for SCC (1.10 to 3.38 ‰), we estimate a $f_{org}$ range of 0.10-0.20 (Table 3), consistent with Bjerrum and Canfield (2004). Assuming an extreme case where OCC contributes only 50 % (i.e. $\lambda$=0.5) of the total inorganic carbon sink, $f_{org}$ remains below 0.23, a scenario deemed unlikely based on field observations. The observed $f_{org}$ range closely matches previous estimates for both the early Archean (Krissansen-Totton et al., 2015) and modern times (Hayes et al., 1999; Berner, 2004), highlighting
the prevalence of carbonates, particularly those in the oceanic crust factory, as primary carbon sinks during early Earth's history.

**Table 3: Parameters used in calculating carbon isotopic mass balance**

| $\delta^{13}C_{in}$ [a] | $\delta^{13}C_{org}$ [a] | $\delta^{13}C_{SCC}$ | $\delta^{13}C_{OCC}$ [b] | $\Delta_S$ | $\Delta_b$ | $\lambda$ | $f_{org}$ |
|---|---|---|---|---|---|---|---|
| -5.0 | -30.0 | 0.0 [a] | -2.37 | -2.4 | -30.0 | 0.95 [a] | 0.099 |
| -5.0 | -30.0 | 0.0 [a] | 0.99 | 1.0 | -30.0 | 0.95 [a] | 0.192 |
| -5.0 | -30.0 | 1.1 [c] | -2.37 | -3.5 | -31.1 | 0.95 [a] | 0.101 |
| -5.0 | -30.0 | 1.1 [c] | 0.99 | -0.1 | -31.1 | 0.95 [a] | 0.193 |

| | | | | | | | |
|---|---|---|---|---|---|---|---|
| -5.0 | -30.0 | 3.4 [d] | -2.37 | -5.8 | -33.4 | 0.95 [a] | 0.105 |
| -5.0 | -30.0 | 3.4 [d] | 0.99 | -2.4 | -33.4 | 0.95 [a] | 0.196 |
| -5.0 | -30.0 | 3.4 [d] | -2.37 | -5.8 | -33.4 | 0.70 | 0.148 |
| -5.0 | -30.0 | 3.4 [d] | 0.99 | -2.4 | -33.4 | 0.70 | 0.212 |
| -5.0 | -30.0 | 3.4 [d] | -2.37 | -5.8 | -33.4 | 0.50 | 0.180 |
| -5.0 | -30.0 | 3.4 [d] | 0.99 | -2.4 | -33.4 | 0.50 | 0.223 |

**Note: superscript a, b, c, and d respectively indicate data from Bjerrum and Canfield (2004), the max/min $\delta^{13}C$ values of carbonates in the oceanic crust factory, the min carbonate $\delta^{13}C$ in the organo-carbonate factory, and the max carbonate $\delta^{13}C$ in the microbial carbonate factory.**

The prevalence of carbonatized greenstones during the early Archean era underscores the significance of oceanic crust factory and OCC (Kitajima et al., 2001; Nakamura and Kato, 2002, 2004; Anhaeusser, 2014; Kasting, 2019; Nutman et al., 2019a). Estimates of $CO_2$ fluxes from ocean to crust during this period, based on carbonate abundance, exceed $3.8 \times 10^{13}$ mol/yr in the early and $1.5 \times 10^{14}$ mol/yr in the middle Archean (Nakamura and Kato, 2004; Shibuya et al., 2012), markedly higher (1-2 orders) than modern fluxes ($1.5–2.4 \times 10^{12}$ mol/yr; Alt and Teagle, 1999). Importantly, prior studies overlooked interstitial carbonates due to quantification difficulties. Our preliminary volumetric assessments suggest a higher proportion of interstitial carbonates than carbonate minerals in basalts (Figs. 3a and 5b), implying a more pivotal role for oceanic crust factory and SCC in shaping the early Archean global carbon cycle than previously thought (Nakamura and Kato, 2004; Shibuya et al., 2012; Coogan and Gillis, 2013).

To assess its significance in the carbon cycle, we estimated carbon flux to the early Archean oceanic crust, focusing on pillow basalts that retain primary interstitial calcite due to uncertainty in altered samples' post-depositional timing. The carbon flux is quantitatively approximated by the product of two key factors: firstly, the carbon incorporated into interstitial calcite, arising from the interaction of seawater, $CO_2$ and basaltic oceanic crust during OCC (see Eq.1), and secondly, the rate of production of altered oceanic crust. It can be written as:

$$F_C = F_{Ca} = R_{cc} \times C_{Ca} \times L_{Ca} / M_{Ca} \qquad (Eq.6)$$

where $F_C$ is the carbon flux into the oceanic crust factory (mol/yr), $F_{Ca}$ is the released calcium flux during OCC (mol/yr), $C_{Ca}$ is the Ca concentration of the oceanic crust, $L_{Ca}$ is the Ca loss during SCC (%), $M_{Ca}$ is the molar mass of Ca (g/mol), and $R_{cc}$ is the production rate of the carbonatized oceanic crust (g/yr). The $R_{cc}$ can be further estimated by Eq.7:

$$R_{cc} = sp \times D \times \rho \qquad (Eq.7)$$

where $sp$ is the spreading rate of oceanic crust (cm$^2$/yr), D is the depth of carbonatized zone (cm), and $\rho$ is the density of oceanic crust (g/cm$^3$).

To enable a comparison with Nakamura and Kato's (2004) results, we used specific parameters: D and *sp* (500 m and $1.8 \times 10^{11}$ cm²/yr, respectively; Nakamura and Kato, 2004), $\rho$ (3.0 g/cm³, akin to modern basalt; Karato, 1983), $C_{Ca}$ (2.61 wt%, matching Primitive Mantle; Palme and O'Neil, 2014), $L_{Ca}$ (22.57 %, see Appendix C), and $M_{Ca}$ (40 g/mol). Our calculation revealed a carbon flux of $3.8 \times 10^{12}$ mol/yr into the oceanic crust factory, one order of magnitude lower than Nakamura and Kato's oceanic crust flux ($3.8 \times 10^{13}$ mol/yr). This difference contrasts with volumetric estimates from thin section analysis, suggesting an issue. Accuracy of µXRF-derived $L_{Ca}$ values, limited by reference materials, may affect representativeness. However, even assuming complete Ca loss from PM during alteration ($L_{Ca}$ of 100%), our oceanic crust factory flux estimate remains lower ($1.8 \times 10^{13}$ mol/yr). This discrepancy hints at an overestimation of average carbon content ($1.4 \times 10^{-3}$ mol/g) in Nakamura and Kato's study, warranting a reassessment of assumptions and methodologies.

Despite uncertainties, the early Archean oceanic crust factory likely served as a significant carbon sink. Integrating modern and Archean data, seafloor weathering carbon flux at 3.46 Ga was estimated at $7.6\text{-}65 \times 10^{12}$ mol/yr (Krissansen-Totton et al., 2018). The same calculation using our data yields a flux range of $0.76 – 6.5 \times 10^{12}$ mol/yr. This range is similar to estimates assuming Archean continental weathering but below those which do no (Krissansen-Totton et al., 2018). Remarkably, modern oceanic crust seems to lack a carbonate factory as observed in the early Archean; carbonate minerals occur primarily in veins, vesicles, and breccias, with greater abundance in older crusts (Gillis et al., 2001; Heft et al., 2008; Coogan and Gillis, 2013). This carbonate factory's carbon flux approximates that in modern oceanic crusts ($1.0\text{-}2.4 \times 10^{12}$ mol/yr; Alt and Teagle, 1999; Rausch, 2012). This underscores the pivotal role of the oceanic crust factory as a major carbon sink on the early Earth's surface, while simultaneously elucidating the previously underestimated contributions of OCC to the carbon cycle and its potent capacity as a climate-modulating buffer during that epoch.

In summary, constraining carbon flux dynamics in the early Archean is challenging due to uncertainties in quantifying carbonate reservoirs. Despite limitations, estimates based on carbon isotope mass balancing across carbonate factories and carbon flux in oceanic crust suggest they were significant carbon sinks. These factories were fundamental to the carbon cycle, acting as buffers that modulated early Earth's climate.

## 6 Conclusion

Paleoarchean rocks in the Pilbara Craton (Western Australia) contain carbonates of various origin. Three carbonate factories are recognized: (i) an oceanic crust factory, (ii) an organo-carbonate factory, and (iii) a microbial carbonate factory. The oceanic crust factory is characterized by carbonates associated with pillowed basalts, which precipitated abiotically on and within basaltic oceanic crust from $CO_2$-enriched seawater and seawater-derived alkaline hydrothermal fluids. The organo-carbonate factory encompasses carbonate that formed via taphonomy-controlled organo-mineralization linked to organic macromolecules (either biotic or abiotic). The microbial carbonate factory includes carbonates formed through mineralization controlled by microbial extracellular polymeric substances (EPS). In case of all three carbonates factories, hydrothermal fluids seem to play also an important role in the formation and preservation of mineral precipitates. Carbon isotope mass balances reveal a $f_{org}$ range of 0.10-0.20, close to what is known from modern Earth. Likewise, the estimated carbon flux into the oceanic

crust factory ($0.76$-$6.5 \times 10^{12}$ mol/yr) is similar to that by oceanic crust carbonatization in the modern ocean. Hence, oceanic-crust related carbon cycling during the early Archean has been somewhat similar to today. Our study underscores the value of Paleoarchean carbonates as geobiological archives and emphasizes their importance as major carbon sinks on the early Earth, highlighting their potential role in modulating the carbon cycle and, consequently, shaping climate variability.

*Data availability.* The data are presented in the manuscript; and can be requested from the corresponding author.

*Author contribution.* Xiang, Reitner and Duda designed the framework and methodology of this study. Reitner and van Zuilen contributed to the fieldwork, sample and data collections. Pack was involved in data interpretation. Xiang is responsible for data collection, analysis and interpretation, and drafting the manuscript. All co-authors have been involved in revising the content of the manuscript, and approved the final manuscript for submission.

*Competing interests*. The authors declare that they have no conflict of interest.

*Acknowledgements.* We thank A. Hackmann and W. Dröse for sample preparation, B. Schmidt, J. Schönig for their help with Raman spectroscopy and µXRF, A. Kronz for carrying out EPMA mappings, and T. Di Rocco, D. Kohl and T. Wasselin for stable isotope measurements, all geoscience faculty of University of Göttingen. M. van Kranendonk (Curtain University Perth, Western Australia) as well as F. Myers and G. Myers acknowledged for their assistance in the field and providing rock material. A. Hickman (Geological Survey of Western Australia) and A. Hofmann (University of Johannesburg, South Africa) thanked
for providing information of the Pilbara Craton and the Barberton Greenstone Belt, respectively. The core library of the Geological Survey of Western Australia, is acknowledged for permission to sample drill core materials from the Pilbara region (approval for P954, 1014, 1091). This study was financially supported by the China Council Scholarship (CSC), the German Research Foundation (DFG) priority program (SPP)1833 "Building a Habitable Earth" (RE 665/42-2; DU 1450/3-1, DU 1450/3-2; TH 713/13-2), the Göttingen Academy of Sciences and Humanities in Lower Saxony, and Leshan Normal University
Scientific Research Start-up Project for Introducing High-level Talents.

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
