# Peer review of "Were early Archean carbonate factories major carbon sinks on the juvenile Earth?"

_EGUsphere, 2024_

## Author Response (AR1)

**Preface**

We are grateful for the invaluable guidance provided by the referees and the associate editor, which has significantly enhanced our manuscript. We have thoroughly revised the Introduction to clarify carbon sinks and cycles during the early Archean era, and introduced a new Discussion section titled "5.3 Carbon Sinks in the Early Archean." This section presents detailed calculations on carbon isotopic mass balance and flux within carbonate factories, addressing the central question posed by our title: " early Archean carbonate factories were major carbon sinks on the juvenile Earth". While these extensive revisions are not fully encapsulated in this response, we have updated the Abstract and Conclusion to reflect these improvements.

Despite some overlapping concerns expressed by the referees, we have opted to address each comment individually to ensure comprehensive coverage of all aspects. However, please note that the reference list for our responses remains consistent throughout.

Additionally, we have made minor adjustments to enhance the accuracy and readability of the manuscript, which are evident in the Author's track-changes file. We sincerely hope that the revisions made to our manuscript effectively convey our message and resonate with all readers.

*RC1: 'Comment on egusphere-2024-1007', Graham Shields, 21 May 2024*

*This fascinating contribution presents new carbon and oxygen stable isotope data, mainly from early Archean (Paleoarchean) carbonates in western Australia, and comes to the conclusion that Paleoarchean carbonates, which the authors divide into three main types, could have been major carbon sinks at the time, thus moderating the global carbon cycle and contributing to early climate regulation. The geochemical data are new, sedimentological context well described, and conclusions of broad interest. I have a few comments for the authors to consider when framing the final version of this thoughtful manuscript.*

1. *RC1: Major carbon sink. The major source of carbon into the Archean exogenic system would have been volcanic outgassing (in the absence of oxidative weathering), while the major sink would have to have been carbonate, just as today. The premise of the paper is that we do not yet know what the carbon sinks would have been at the time, but I would ask the authors what other sinks might there have been, because they do not specify the alternatives anywhere and do not provide a conceptual box model of the Archean carbon cycle, which would have been useful. If they draw the same conclusion as me, that carbonate minerals must have been the dominant sink, then I suggest rewording the title, conclusions and other relevant sections throughout the manuscript. Even accepted that it might not have been the dominant sink, which I very much doubt, is it likely that carbonate was ever not a "major" sink as implied by the title? In this regard, the final sentence of the Abstract is also rather weak.*

   R: Thank you very much for offering us these valuable and helpful suggestions. Firstly, we fully concur that carbonate minerals likely served as the major carbon sink during that period, which is bolstered by our calculations based on carbon isotopic mass balances and fluxes, incorporating the esteemed referees' feedback. However, despite having adjusted the title in our previous response, we maintain that the original title better encapsulates our intentions: to captivate readers and, crucially, foster further research endeavors in this vital area. Our rationale stems from the challenges encountered during calculations, where numerous poorly constrained parameters hindered the attainment of definitive results.

   To address the issues in RC1, we have undertaken a comprehensive overhaul. We have rewritten the Introduction to clarify the distinction between modern and early Archean carbon sinks, appended a fresh Discussion section titled "5.3 Carbon Sinks in the Early Archean" and refined both the Abstract and Conclusion. These cumulative revisions converge on the conclusion that "early Archean carbonate factories were major carbon sinks on the juvenile Earth". For brevity and ease of reference, the revisions of the Abstract and Conclusion are outlined below:

a) Abstract: "… Regardless of the formation pathway, Paleoarchean carbonates might have been major carbon sinks on the early Earth, as additionally suggested by carbon isotope mass balances indicating a carbon flux of 0.76-6.5 × $10^{12}$ mol/year. Accordingly, these carbonates may have played an important role in modulating the carbon cycle and, hence, climate variability, on the early Earth."

b) Conclusion: "…Carbon isotope mass balances reveal a $f_{org}$ range of 0.10-0.20, close to what is known from modern Earth. Likewise, the estimated carbon flux into the oceanic crust factory (0.76-6.5 × $10^{12}$ mol/yr) is similar to that by oceanic crust carbonatization in the modern ocean. Hence, oceanic-crust related carbon cycling during the early Archean has been somewhat similar to today. Our study underscores the value of Paleoarchean carbonates as geobiological archives and emphasizes their importance as major carbon sinks on the early Earth, highlighting their potential role in modulating the carbon cycle and, consequently, shaping climate variability."

2. *I would appreciate more quantification and quantified comparisons with the modern. For example, on line 178 "highly carbonatized" could be quantified, while what I really want to know as reader is how this compares with ocean basalts today? Do your observations match the constraints mentioned later from the literature on lines 514-516, for instance?*

R: Thank you for bringing those issues to our attention. We describe "Spherulitic and variolitic zones in the basalts are highly carbonatized", distinct from other basaltic components. This observation stems primarily from elemental mapping via micro-XRF analysis of sample thin sections, as illustrated in Figures 3b and S1a-c.

To further engage our readers, we have undertaken the following additional analyses:

a) We have quantified the mass changes in Si and Ca during oceanic crust carbonatization, leveraging micro-XRF point spectrum data from various basaltic regions (see Appendix C).

b) We have estimated the carbon flux into the oceanic crust, assuming complete transformation of lost Ca into interstitial calcite. Applying parameters from Nakamura and Kato (2004) and Krissansen-Totton et al. (2018), we arrive at carbon flux estimates of 3.8 × $10^{12}$ mol/yr and 0.76 – 6.5 × $10^{12}$ mol/yr, respectively, which align with modern oceanic crust estimates (1.0-2.4 × $10^{12}$ mol/yr; Alt and Teagle, 1999; Rausch, 2012). Notably, these prior estimates overlooked interstitial carbonates, which, though challenging to measure, appear abundant compared to carbonate minerals within basalts (Figures 2, 3, 5). However, our carbon flux estimate falls one order of magnitude below Nakamura and Kato's estimate (3.8 × $10^{13}$ mol/yr), highlighting limitations in current methodologies for accurately determining carbon sequestration during oceanic crust carbonation.

3. *Carbonate mineralogy. Wherever possible I would encourage the authors to specify the mineralogy, e.g. in Table 1 or line 233. This is particularly important where the stable isotope data are outlined / discussed, as different minerals fractionate differently. On line 225, and elsewhere, we learn that the primary mineralogy of interstitial carbonates was calcite, and yet the crystals are described as being "acicular, a habit more commonly associated with aragonite. Can the authors rule out aragonite as the precursor carbonate mineral?*

R: Thank you for your suggestion and pointing the problem. Following your advice, we will add a column in Table 1 to specify the mineralogy. For the question, we do not think of the presence of aragonite as precursor. Even in modern oceanic crust, aragonite is not so pervasive, for that their precipitation may be determined by several factors such as Mg/Ca ratio of parenting fluid and nucleation template. In our work, we have done XRD and Raman analyses, and had the geochemical compositions, which indicate they are now low-Mg calcite. The habit of acicular crystal fan can also be found in calcite, and there should still be some evidence preserved if aragonite was the precursor due to the rather low diagenetic overprint. However, we did not find any convincing evidence to prove "the primary mineralogy of interstitial carbonates was aragonite". Therefore, we chose to describe and believe what we see today.

4. *Other geochemical data. On lines 368-374, we learn about other pertinent geochemical data, such as strontium isotopes, which come from a doctoral thesis. This section comes rather out-of-the-blue, and it is not apparent why these data, which are evidently from the same project, have not been presented more fully. As they are key to the interpretation, I'd recommend that the reader be told more relevant methodological and contextual details about these analyses.*

R: Thank you for pointing out this issue. As per your mention, this work, the Sr isotope data and the aforementioned data of Apex pillow basalts are from the same project that will be in three publications.
Notably, the Sr isotope data are currently unpublished but vital to our ongoing research endeavors. In light of your thoughtful suggestion, we have included supplementary information (Appendix B) that outlines the methodological nuances and contextual background of these analyses. Additionally, we've provided a table showcasing select $^{87}Sr/^{86}Sr$ ratios within the basalt-carbonate system. However, for a more exhaustive account, we warmly recommend referring to Xiang's 2023 Ph.D. thesis, "Carbonate factories in the early Archean and their geobiological impacts," which delves deeper into these topics. Chapters 2 through 4 of our work offer detailed insights into each of the three publications, including intriguing applications of our research findings. We hope this additional context and guidance will be of value to our readers.

5. *Carbon (isotope) mass balance. Carbon sinks and the global carbon cycle do not get enough attention until the end of the paper, but could have come already in the Introduction considering their importance to the "take-home" message. In this regard, the authors need to use their new data and compiled literature data to constrain the isotopic value of the carbonate sink at that time, and carry out a simple isotope mass balance calculation. The isotopic composition of the other sink, organic matter, can be estimated from the literature, but it is not mentioned in the paper. These values allow a very simple isotope mass balance to be proposed if we can assume a likely input value or range of input values (this is mentioned in the paper). As far as I can tell, the conclusion from such a simple mass balance would confirm that the sub-seafloor calcite carbon sink was likely the dominant carbon sink of the Paleoarchean (lines 511-512). A pertinent paper in this regard would be Mills et al (2016) Proterozoic oxygen rise linked to shifting balance between seafloor and terrestrial weathering, in PNAS, 11, 9073-9078, but there are other relevant papers not mentioned here that pertain to the carbon budget on the early Earth, e.g. Canfield (2021) Carbon cycle evolution before and after the great oxidation of the atmosphere. American journal of Science, 321, 297-331.*

R: Thank you for your kind suggestions and recommending nice references. In the response to your first comment, we have refined the Introduction to introduce carbon sinks and the carbon cycle earlier, emphasizing their pivotal role in moderating climate. Specifically, within the "5.3 Carbon Sinks in the Early Archean" section, we have conducted carbon isotope mass balance calculations adhering to the methodology outlined in Bjerrum and Canfield (2004), which aligns well with our research context, rather than Canfield (2021). These calculations, coupled with our quantification of carbon flux into the oceanic crust factory, underscore the preeminence of carbonates, particularly those formed in the oceanic crust factory, as the dominant carbon sinks during the early stages of Earth's history.

6. *English formulation and other minor issues:*
(1) *Line 20 (Abstract): "Interspaces between pillowed basalts" – this is not the same as interstitial carbonate, and needs rewording for clarity.*

R: Thanks for pointing this problem. Following your advance, the sentence (Line 20) "The oceanic crust factory is characterized by carbonates formed in interspaces between pillowed basalts ('interstitial carbonates')" has been reworded into "The oceanic crust factory is characterized by carbonates formed in void spaces of basalt pillows (referred to as "interstitial carbonates" in this work)".

(2) *Substitution or modification of words. We organized some comments to answer here:*
   a. *Line 23 (Abstract): "formed taphonomically" can be omitted here and elsewhere, as it is not clear what extra meaning this adds, as this term usually refers to fossil preservation.*
   b. *Line 74: I suggest omitting "comprehensively" here. Likewise, on line 149 "high-*

*precision" seems unnecessary.*

c. *Line 158: "into", not "in".*
d. *Line 417: "consummated" probably needs a different word?*
e. *Line 461: "complexation" – for me, this term means something else, as in complexed CaCO30.*
f. *Line 468: "higher", not "heavier".*

R: Thank you for pointing out these problems. Following your advice, we will omit "taphonomically" on Line 23, "comprehensively" on Line 74, "high-precision" on Line 149, and correct "in" on Line 158 with "into", "consummated" on Line 417 with "confirmed", "complexation" on Line 461 with "combination", "heavier" on Line 468 with "higher". After the corrections, the sentences will be presented as below:

a. Line 23: "The close association with organic matter suggests that the carbonates formed via organo-mineralization, that is, linked to organic macromolecules (either biotic or abiotic) which provided nucleation sites for carbonate crystal growth."
b. Line 74: "This study investigates early Archean carbonates in the EPT, including interstitial carbonates associated with basalts, carbonate stromatolites and other sedimentary carbonates."
   Line 149: "Additionally, some carbonate facies, including carbonate veinlets and carbonate inclusions, were extracted using a drill from individual mineral phases from polished rock slabs."
c. Line 158: "The host basalts are pillow-shaped, internally subdivided into more crystalline interiors and quenched glassy rims, and commonly locally cut by tectonic fractures (Fig. 2)."
d. Line 417: "To distinguish minerals formed through mineralization linked to organic matrices and compounds from those whose formation is induced by living organisms, the terms "organomineral" and "organo-mineralization" were introduced at the 7th International Symposium on Biomineralization in 1995 and further developed in the following decade (…), before being finally confirmed in following studies (…)."
e. Line 461: "Certain functional groups of organic substances in the EPS (e.g. Asp- and Glu-rich macromolecules) efficiently bind and sequester divalent cations such as $Ca^{2+}$ and $Mg^{2+}$, thereby inhibiting their combination with carbonate anions and subsequent precipitation (Reitner et al. 1995a, b, c)."
f. Line 468: "More specifically, $\delta^{13}C$ values of carbonates from SPF stromatolites (3.08 ± 0.30 ‰ on average) are higher than those of the interstitial carbonates (0.22 ± 0.98 ‰ on average) and the sedimentary carbonates (1.85 ± 0.48‰ on average)."

7. *Lines 194-195: Here, and elsewhere I would have appreciated a chemical equation to illustrate the process being described.*

R: Thank you for your kind suggestion. Following your advice, we have made corrections here to illustrate it.

Firstly, we have replaced some words "Spherulitic and variolitic zones in the basalts are highly carbonatized, with carbonate minerals being particularly prominent in variolites and concentric syngenetic veins. Notably, elemental distributions in basalts do not seem to relate to the degree of weathering (e.g. in sample A22 of the Apex Basalt; Fig. 3b) and hence might be pristine. Except for the devitrified volcanic glass, Si is rich in the interior of the pillow basalt but rare in the zone of spherulites and variolites, which are dominated by calcite" in the end of the 2$^{nd}$ paragraph in Section 4.1.1, with "Carbonate minerals are particularly prominent in voids, veins and variolites within alteration zones, as illustrated by μXRF element overlay images of Si, Ca and Mg (Fig. 3b) as well as by calculated Ca mass changes (Fig. S14b; Appendix C). Except for the devitrified volcanic glass, Si is rich in the interior of the pillow basalt but rare in the alteration zones (Figs. 3b, S14), implying a Si loss during basalt carbonatization. Si yielded during this process was likely enriched in fluids, resulting in chert cementation of interstitial carbonates (Fig. 4). The process can be summarized as follows (Eq.1; note that "$CaSiO_3$" refers to calcium silicate minerals):

$$"CaSiO_3" + CO_2 + H_2O \rightarrow "CaSiO_3" + H_2CO_3 \rightarrow CaCO_3 + SiO_2 + H_2O \qquad (Eq.\ 1)"$$

Secondly, words on Lines 194-195 in the caption of Fig. 3 have been now "In addition, the quenched margin of the basalt seems to relatively depleted in Si as compared to the core, implying a loss of Si during carbonatization processes (see Eq.1)."

8.  *Lines 244-245 – the evidence for dolomitization is given as the presence of ankerite. I find this confusing. Isn't this "ankeritization"?*

R: Thank you for pointing this issue. As you said, this is ankeritization. It was me who insisted to use "dolomitization", while some of the coauthors thought of "ankeritization". I preferred to use the usual term, because dolomitization and recrystallization are two common processes during diagenetic alteration, and ankerite [Ca(Fe,Mg,Mn)(CO$_3$)$_2$] is an Fe- and Mn-bearing dolomite. However, considering your advice and the truth that ankerite is a characteristic mineral which is only found in interstitial carbonates, we have corrected "dolomitization" with "ankeritization" in the relevant sentences, e.g. on Lines 238 and 244.

9.  *Lines 250-251 – sentence needs rewording for clarity. Likewise, lines 438-439, 463-465, 486-487.*

R: Thank you for your kind suggestions. Following your advice, we will make corrections as follows:
a.  Lines 250-251: we will revise the paragraph into "The secondary carbonates are either Mn- or Sr-enriched (Fig. 5), indicating the influence of at least two diagenetic fluids during later alteration. Secondary Mn-enriched carbonates include recrystallized

interstitial calcites and ankerites as well as calcite cements within basalt fractures. Notably, the degree of Mn-enrichment in interstitial ankerites varies, with those formed through recrystallization and neomorphism or closer to the basaltic parts being relatively more enriched (Fig. 5a). At the same time, calcites overgrowing interstitial ankerites and, even more so, within fractures are enriched in Sr (Fig. 5)."

b.  Lines 438-439: "Some EPT bedded sedimentary carbonates (except the Dresser bedded carbonates) show an average $\delta^{13}C$ value of $1.85 \pm 0.48$‰ (Fig. 11). These is consistent with other reports on Strelley Pool stromatolites (Lindsay et al., 2005; Flannery et al., 2018) and within the range of modern seawater (Kroopnick, 1980; Tan, 1988), reflecting their formation in marine environments."

Lines 463-465: "This process is somewhat similar to organo-mineralization, which involves a mineralization of organic matrices and compounds decoupled from the source organisms or of abiotic origin (Trichet and Défarge, 1995; Défarge et al., 2009; Défarge, 2011)."

c.  Lines 486-487: "Carbonates in this carbonate factory precipitated from $CO_2$-rich seawater-derived hydrothermal fluids characterized by a high alkalinity and high cation loads."

10.  *Table 2 – How are evaporite minerals identified in this study? And which minerals are these?*

R: Thanks for pointing this problem out. We assume there should be some evaporite minerals in organo-carbonate factory and microbial carbonate factory due to their deposition environments (supported by references and this work) and mineral morphologies. For example, the organo-carbonate factory could occur on land, as showed in the case of the DB bedded carbonates, in a hydrothermal pond. "The clusters of radiating calcite crystals at the base of each carbonate-chert layer (Figs. 6f, 8a), which were initially proposed to be gypsum or aragonite (Runnegar et al., 2001; Van Kranendonk et al., 2008; Otálora et al., 2018), are likely indicative of evaporitic conditions" (Lines 442-445). In the microbial carbonate factory, we ascribed the formation of carbonate fans beneath the SPF stromatolites to be evaporation, considering their morphology and the depositional environment of the SPF stromatolites. Therefore, the evaporate minerals in our study are carbonate minerals (calcite and dolomite).

However, it appears that we have inadvertently omitted some crucial messages for our readers. To solve this problem, we have made the following corrections:

a.  We will supplement the information of carbonate fans beneath the SPF stromatolites in the end of the 2nd paragraph in Section 4.3: "The stromatolites occur atop large, chert cemented carbonate fans (~ 40 cm) situated on a chert layer (Fig. 9). The carbonate fans encompass fusiform dolomite aggregations (Fig. 9f)."

b.  We will add the introduction of depositional environments in the first paragraph in Section 5.2 from Line 485 (underlined): "The oceanic crust factory includes abiotically formed carbonates such as Mn- or Sr-enriched calcite and ankerite that are associated

with pillow basalts within the upper oceanic crust. Carbonates in this carbonate factory precipitated from $CO_2$-rich seawater-derived hydrothermal fluids characterized by a high alkalinity and high cation loads. The organo-carbonate factory is dominated by authigenic carbonates formed through taphonomy-controlled organo-mineralization (i.e. organomicrites). Importantly, and in contrast to the microbial carbonate factory, the involved organic matter can be of either biological or abiotic origin. For this reason, precipitates assigned to this carbonate factory formed in various environments, ranging from shallow marine to terrestrial settings. The microbial carbonate factory is somewhat similar to the organo-carbonate factory, but specifically refers to EPS-controlled carbonate precipitation, that is, mineralization of biologically derived organic substances. However, as in case of the organo-carbonate factory, organomicrite is formed as a typical product. Since this carbonate factory is directly linked to biological activity, the assigned precipitates typically occur in the photic, relatively restricted, shallow marine environments like lagoons. Given that most of these carbonates formed in shallow-water environments under anoxic conditions, anoxygenic phototrophs appear a plausible source of biological organic matter, but this remains to be tested in future studies."

*RC2: 'Comment on egusphere-2024-1007', Anonymous Referee #2, 28 May 2024*

*In this manuscript, the authors characterize three different sources of carbonates commonly found in the East Pilbara Terrain (Australia). They describe and differentiate the potential origin of said carbonates in three different marine carbonate factories, as reflected in their distinct d13C signal, these being: 1. The oceanic crust (samples obtained from interstitial carbonate within basalts); 2. Organo-carbonates (sedimentary carbonates form through mediation of an organic component); and 3. Microbial carbonate (stromatolites).*

*Although I enjoyed reading this manuscript, and it made me think about the carbon cycle from a different perspective that I normally would, I recommend some changes below to be carried out before its publication.*

1. *First, and foremost, either the title needs to be changed, or the discussion needs to be rewritten. As it stands, it is unclear to me that this manuscript answers their own title "Were early Archean carbonate factories major carbon sinks on the juvenile Earth?". The last sentence in the manuscript says "[...] Paleoarchean carbonates might have been major carbon sinks at the time of formation [...]". So, were they major carbon sinks? There is no mention of any quantification of the discussed carbon sink, nor mention of any other potential carbon sinks during this geological period. After reading this manuscript, I cannot answer the question that the authors are postulating. Considering this journal, it is likely that non-experts in Archean carbonates will read this paper, and a further expansion on the carbon cycle at the time, literature looking at carbon sinks in this time and how this work fits with the literature are needed.*

   R: Thank you for your insightful feedback and generous suggestions. We acknowledge that initially, we had assumed the early Archean carbonates to be the prevalent carbon sinks during that era. Seeking to engage readers, we attempted to frame the title as a question, but inadvertently caused confusion, which was never our intention. To rectify this and enhance clarity, we've embarked on a thorough revision. We've rewritten the Introduction to sharply distinguish between modern and early Archean carbon sinks. Additionally, we've appended a new section titled "5.3 Carbon Sinks in the Early Archean", which delves into carbon isotopic mass balances and flux calculations, solidifying our stance that carbonate factories served as major carbon sinks. Consequently, we've refined both the Abstract and Conclusion to align with these clarifications. We sincerely hope these improvements facilitate a clearer understanding of our work for our esteemed readers.

2. *Also, the introduction needs to be organized in a different way in order to bring to the front the research question the authors are trying to answer, which is not clear until the very end of the introduction. As before, given the varied potential readership of the journal, the point of why to investigate Archean carbonates, and the ones in the EPT in particular, needs to be specified early in the introduction. For example: 1) The carbon cycle and carbon sinks and their importance for the climate system during the Precambrian. There is no mention of this until the very end of the Introduction (Line 79), and then again nothing until the very*

*end of the discussion (Line 505), this needs to come up earlier, and expanded. 2) Then, as per lines 51 – 54, carbonate formations in this period are poorly constrained, especially in the early Archean. 3) This sets the framework to drive the reader into why more effort needs to be paid, and why you do the work you do. This is only an example of a potential structure that could highlight the importance of this work. Essentially, the introduction needs to be "punchier".*

R: Thank you for bringing this matter to our attention. Taking heed of your suggestions along with RC1, we have revised the Introduction comprehensively. To facilitate a clear and concise overview, we have chosen to include the opening and closing sentences of each paragraph, thus illustrating the structure succinctly for your easy reference:

Paragraph 1[st]: "Biogeochemical carbon cycling plays a crucial role in maintaining the stability of modern Earth's climate system (Ciais et al., 2013). …Understanding the formation and evolution of carbonate factories— that is, conceptual models encompassing carbonate production and associated processes at various scales, from local precipitation to global sedimentation (Schlager, 2000; Schrag et al., 2013; Reijmer, 2021)—is therefore essential for comprehending the dynamics of the carbon cycle and its implications for climate change."

Paragraph 2[nd]: "Throughout most of Earth's history, carbonate precipitation has been closely linked to biological processes, ranging from direct to indirect precipitation (that is, biologically controlled vs induced) (Flügel, 2010). …Hence, geological and biological key-processes – and by extension Archean carbonate factories – must have been very different as compared to any later stage in Earth's history."

Paragraph 3[rd]: One of the most important early Earth records is the ca. 4.0–3.6 Ga Isua Supracrustal Belt (ISB; West Greenland), albeit being highly metamorphic (amphibolite facies: Nutman et al., 2019a, b). …With regard to carbonates, most studies have focused on occurrences associated with microbial facies in the ~3.4 Ga Strelley Pool Formation, yet those constitute a minor component within the EPT lithostratigraphy (Van Kranendonk et al., 2007b)."

Paragraph 4[th]: "Basaltic rocks of the EPT show evidence of pervasive carbonatization and silicification associated with hydrothermal processes in subaquatic environments (Kitajima et al., 2001; Nakamura and Kato, 2002, 2004; Terabayashi et al., 2003). …Nevertheless, the informative value of these approaches is limited due to the scarcity of theoretical frameworks and geological baseline data that would allow to constrain such additional carbon sinks, thereby impeding a comprehensive understanding of carbon cycle dynamics over geological time scales."

Paragraph 5[th]: "To address this issue, this study investigates early Archean carbonates in the EPT, …and indicate that they might play a significant role in the early global carbon cycle and, hence, climate system."

3. *Finally, in the results (sections 4.1.2 and 4.1.3) as data is presented on primary and secondary carbonate facies and different minerals found. Is it possible to include any sort of quantification/percentage on the presence of minerals on the analyzed samples? This would give a more complete characterization of the interstitial carbonates, the diagenetic*

*processes and their stable isotopic composition.*

R: Thank you for highlighting this issue. We conducted XRD measurements on mineral percentages in select samples, yet found them less insightful than direct observation of thin sections for elucidating carbonate diagenetic processes. Specifically, XRD could not discern between primary calcite and various recrystallized forms. Consequently, we opted for in-situ analyses. When necessary for stable isotope analysis, we carefully sampled individual mineral facies using micro-drilling, guided by thin section observations (yielding insufficient material for XRD). Regarding Table 1 data, we suspect most samples were analyzed based on a single carbonate facies, except interstitial ankerite samples from Mount Ada Basalt, which were complicated by calcite overgrowth (refer to Fig. 4i). Currently, we lack a robust method to quantify mineral percentages in our samples. Furthermore, quantifying interstitial carbonates, crucial for estimating carbon fluxes into oceanic crust, poses a challenge. In summary, exploring effective in-situ techniques for quantifying mineral percentages remains an intriguing area for future research.

*Minor comments:*

4. *Substitution or modification of words. We organized some minor comments to answer here:*
   a. *Line 92: Repeated "consisting" word twice within a sentence. Rewrite for clarity.*
   b. *Line 146: Change for "were cleaned three times in ethanol using an ultrasonic bath", or similar. "Using ultrasound" might be confusing.*
   c. *Line 147: No need to "gently" dry samples. You can delete this adjective.*
   d. *Caption in Figure 7/Line 291: "(c) Raman spectra for spots in (a) and (c)". It should be "in (a) and (b)".*
   e. *Line 301: "Notably, it occurs between a unit consisting of sulfidic stromatolites […]" Remove consisting.*
   f. *Line 440: "[…], which in good accordance with d13C signatures […]". Remove word "which".*
   g. *Lines 451 – 452: "Biogenicity" repeated twice in the same sentence. Rewrite for clarity.*

R: Thank you for kindly pointing out these mistakes. Following your advice, we will omit the second "consisting" on Line 92, "consisting" on Line 301, "which" on Line 440, and reword the sentence on Line 146-147, "(a) and (c)" with "(a) and (b)" on Line 291, the "biogenicity" with "that" on Line 452. For your convenience to review, the sentences after the correction will be presented as below:

a. Line 92: "A characteristic feature of the EPT is the so-called dome-and-keel structure, consisting of a central nucleus of the 3459 ± 18 Ma North Pole Monzogranite ("North Pole Dome") surrounded by little-deformed, predominantly mafic volcanic rocks of the Warrawoona Group and Kelly Group (Hickman and Van Kranendonk, 2012a) (Fig. 1)."

b-c. Line 146-147: "The sample chips were cleaned three times in ethanol using an ultrasonic bath and dried at room temperature before being crushed into small pieces."

d. Line 291: "(c) Raman spectra for spots in (a) and (b), supporting the presence of ankerite and organic matter."

e. Line 301: "Notably, it occurs between a unit of sulfidic stromatolites and bladed barite below, and wave rippled volcanoclastic sediments above (Fig. 6e)."

f. Line 440: "At the same time, $\delta^{13}C$ values of the Dresser bedded carbonates are relatively depleted (-5.72 ±1.36 ‰ on average), in good accordance with $\delta^{13}C$ signatures of carbonatites (-4.99 ± 1.22 ‰ on average) (Fig.11), indicating hydrothermal admixture of mantle-derived carbon."

g. Line 451: "Although the biogenicity of early Archean stromatolites is commonly controversial, that of stromatolites from the Dresser Formation and the SPF has been widely accepted (…)"

5. *Lines 125 – 128: "For comparison, we additionally analyzed [...]". What is the purpose of this comparison? Include this info in this paragraph, it will make it easier for the reader to understand why you included these samples at all.*

   R: Thank you for pointing this out. Following your advice, we will reword the sentence: "In order to better understand depositional environments of the studied EPT carbonates, we additionally analyzed the carbon and oxygen isotopic compositions of diverse reference materials for comparison. They include carbonate inclusions in black barites…"

6. *Lines 177 – 181: These sentences are repetitive and need to be reworded/rewritten.*
   a. *"Spherulitic and variolitic zones in the basalts are highly carbonatized, with carbonate minerals being particularly prominent in variolites and concentric syngenetic veins." – What does highly carbonatized mean? First and second part of the sentence is saying the same thing?*

   R: Thank you for pointing those problems. "Highly carbonatized" was used to describe spherulitic and variolitic zones, compared to other basaltic parts. The second part of the sentence is an explanation for the first part. To clarify, we have revised the sentence regarding "highly carbonatized" zones, now stating that "Carbonate minerals are particularly prominent in voids, veins and variolites within alteration zones, …" Furthermore, in response to RC1's guidance and your valuable suggestion, we have incorporated a quantification of the carbonatization degree, specifically Ca mass changes, into our supplementary materials, which can be found in Appendix C.

b. *"Notably, elemental distributions in basalts do not seem to relate to the degree of weathering (e.g. in sample A22 of the Apex Basalt; Fig. 3b) and hence might be pristine." – I do not know what do you mean by this. What would be different in the data if elemental distribution related to weathering? Also, likely need some references to justify why does not relate to degree of weathering. And then, this would be more fitting included in the discussion.*

R: Thank you for bringing this to our attention. Our initial mention of the weathered basalt part in Fig. 3a stemmed from our observation and prior knowledge that elements like Ca, Mg, and Al can be easily lost during weathering. While we anticipated this might affect elemental distributions, we now recognize that the weathered area primarily comprises silicates and quartz, which are more resilient to weathering. Given its potential for confusion and lack of direct relevance to our current work, we have decided to omit that sentence for clarity and conciseness.

c. *"Except for the devitrified volcanic glass, Si is rich in the interior of the pillow basalt but rare in the zone of spherulites and variolites, which are dominated by calcite" – You are mentioning again that calcite is predominant in spherulites and variolites. Which you mentioned two sentences ago.*

R: Thank you for pointing this issue. Taking into consideration all relevant corrections, we have revised the paragraph as follows:

"Although the host basalts show secondary mineral assemblages indicative of greenschist metamorphism (calcite + chlorite + anatase + quartz ± pyrite), phenocrysts (i.e., plagioclase and pyroxene) can still be recognized in the basalt interior of the well-preserved samples, e.g. A22 from the Apex Basalt (Figs. 3a, 4a). Notably, the well-preserved basalts exhibit concentric green ophitic-holohyaline interiors and yellow-green quenched margins. In the margins, the size and density of ovoid spherulites and variolites (amygdules) decrease outwards, merging into the glassy zone (Fig. S1a-c). Carbonate minerals are particularly prominent in voids, veins and variolites within alteration zones, as illustrated by μXRF element overlay images of Si, Ca and Mg (Fig. 3b) as well as by calculated Ca mass changes (Fig. S14b; Appendix C). Except for the devitrified volcanic glass, Si is rich in the interior of the pillow basalt but rare in the alteration zones (Figs. 3b, S14), implying a Si loss during basalt carbonatization. Si yielded during this process was likely enriched in fluids, resulting in chert cementation of interstitial carbonates (Fig. 4). The process can be summarized as follows (Eq.1; note that $"CaSiO_3"$ refers to calcium silicate minerals):

$$"CaSiO_3" + CO_2 + H_2O \rightarrow "CaSiO_3" + H_2CO_3 \rightarrow CaCO_3 + SiO_2 + H_2O \qquad (Eq. 1)"$$

7. *Line 368: On the sentence referring to Sr isotopes. I had to dig where this information was coming from. I was confused as to where the calcite Sr signal (0.700596) was coming from. Is it the same samples as this manuscript? Different samples but same location? Data from a completely different location? And why is typical of Archean seawater? I think this needs to be expanded, as it is in strong support of your stable isotopic data. There is no other mention of Sr isotopes in the rest of the manuscript, or other supporting literature. Also repeated in Line 393.*

R: Thank you for pointing out this issue. Our investigation into the EPT carbonate rocks has encompassed sedimentology, mineralogy, and geochemical compositions, resulting in two forthcoming articles. Utilizing the classification of carbonate facies, both primary and secondary, we've selectively analyzed samples for their geochemical makeup, specifically REEs and Sr isotopes, through acid digestion. This preparatory work has yielded two sets of data: one comprehensive system encompassing mineralogy, elemental, and isotopic compositions (C, O, Sr) for select samples, and another focusing on mineralogy, C, and O isotopic compositions for others (as detailed in Table 1).

Notably, the calcite Sr signal (0.700596) sourced from fracture-filling calcite (D-2-W) aligns with Archean seawater characteristics, corroborated by the REE+Y pattern (low REE concentrations, no Eu anomaly, positive Y anomaly etc) and its $^{87}Sr/^{86}Sr$ ratio, which closely mirrors the lowest recorded for barites from the Dresser Formation (0.700502 in McCulloch, 1994; 0.700447 in Chen et al., 2022). We will delve deeper into this in a separate article.

Parallelly, we've employed the same methodology to examine pillow basalts from Apex Basalt, preserving primary interstitial carbonates, which is currently under preparation. Our Rb-Sr errorchron analysis of these basalts yields a whole rock value of 0.706337 ± 0.000078, linking it to our interstitial carbonate studies. This connection is evident in our discussion (Lines 393-395), highlighting the interplay of fluids, as evidenced by the $^{87}Sr/^{86}Sr$ ratios spanning from early Archean seawater to Apex basalt.

While these findings are currently unpublished and integral to our ongoing work, we acknowledge your feedback and readers' interest. Therefore, we have provided a supplementary material outlining the methodologies, context, and a table summarizing key $^{87}Sr/^{86}Sr$ ratios across the basalt-carbonate system (see Appendix C). For a more comprehensive understanding, we encourage readers to explore Xiang's 2023 Ph.D. thesis, "Carbonate factories in the early Archean and their geobiological impacts," where they will discover fascinating applications of our research.

8. *Line 438: "δ13C signatures of some EPT bedded sedimentary carbonates except the Dresser bedded carbonates (1.85 ± 0.48‰ on average) are generally in line with a formation in marine environments". As it reads, seems like the 1.85% value is that of the Dresser carbonates.*

R: Thank you for pointing out this problem. To solve it, we have revised this sentence for clarity: "Some EPT bedded sedimentary carbonates (except the Dresser bedded carbonates) show an average $\delta^{13}$C value of 1.85 ± 0.48‰ (Fig. 11). These is consistent with other reports on Strelley Pool stromatolites (Lindsay et al., 2005; Flannery et al., 2018) and within the range of modern seawater (Kroopnick, 1980; Tan, 1988), reflecting their formation in marine environments."

9. *Line 458 – 459: "In any case, extracellular polymeric substances (EPS) secreted by microorganisms, amongst others to cope with environmental stressors, play a key-role in mineralization". The part of "among others to cope with environmental stressors" is confusing and distracting. Although I understand what the authors are inferring to, I would rewrite it, and there is no need to dig into the secretion of EPSs to cope with environmental stressors, which made me think about whether the discussion was heading that way (environmental stressors of the stromatolites in this location and period time). For example, you can entirely delete "among others to cope with environmental stressors" and the sentences would still perfectly fit into the discussion.*

R: Thank you for your kind suggestion. We have made a correction following your advice: "In any case, extracellular polymeric substances (EPS) secreted by microorganisms play a key-role in mineralization (Decho, 2011; see in Fig. 10)" (Line 458).

10. *Line 469: "(~0.22 -1.85 ‰ on average; Fig. 11)". These values are confusing, I understand they come from interstitial carbonates (0.22) and sedimentary carbonates (1.85) but all results through the manuscript are given as "X ±Y ‰", so these are not consistent. I had to go back to results to check that 1.85 was from the sedimentary carbonates, and not a ±‰.*

R: Thank you for pointing this problem. Following your advice, we have revised the sentence into "More specifically, $\delta^{13}$C values of carbonates from SPF stromatolites (3.08 ± 0.30 ‰ on average) are higher than those of the interstitial carbonates (0.22 ± 0.98 ‰ on average) and the sedimentary carbonates (1.85 ± 0.48‰ on average)".

11. *Line 470 – 471: "This difference is well in line with a sequestration of 12C by photoautotrophic microorganisms in the microbial mats, resulting in an enrichment of 13C in the environment and, consequently, in the carbonate". This sentence needs a reference.*

R: Thank you for your suggestion. Following your advice, we have added a reference "(Arp et al., 2011)" in the end of this sentence. For your convenience, we attach the reference

here: Arp, G., Helms, G., Karlinska, K., Schumann, G., Reimer, A., Reitner, J., & Trichet, J. (2011). Photosynthesis versus Exopolymer Degradation in the Formation of Microbialites on the Atoll of Kiritimati, Republic of Kiribati, Central Pacific. Geomicrobiology Journal, 29(1), 29–65. https://doi.org/10.1080/01490451.2010.521436.

12. *Figure 3: In 3b, Close-up area imaged under white light seems narrower than the uXRM maps. Can this be amended so they reflect the exact same area?*

R: Thank you for pointing this problem out. It is a display mistake happened when we converted the MS word into pdf. We are sorry for not observing it. We will make a correction and check for the next submission.

13. *Figure 7: Is the scale bar in 7d and 7e both 200um? If 7e is a close-up of 7d, shouldn't the scale bar be wider in the close-up? All scale bars in the images are 200um except that in 7f? Clarify.*

R: Thank you for pointing these problems. The scale bar in 7d and 7e are both 200um, however, 7e is not a close-up of 7d (they are two areas). Besides, 7f and 7g share the same scale bar of 5 um. To eliminate any potential misinterpretation, we have undertaken the following revisions: Firstly, we have aligned the crystals in 7e for clarity and revised the caption to "(e) Close-up view of calcite rhombs showing Mn-enriched dolomite particles within the calcite crust (some crystals are indicated by dotted lines)." Secondly, we have included a scale bar in 7g and revised the caption on Line 295 to read: "The scale bars in (f) and (g) indicate 5 μm, whereas all other figures utilize a scale of 200 μm." These adjustments aim to enhance the accuracy and comprehensibility of our presentation.

14. *Figure 11: This is only a suggestion, but it took me a while to digest the legend and what each symbols is. I would organize the legend grouping by the three carbonate factories discussed in the manuscript. So first group, Oceanic Crust, and include below the interstitial carbonate, veinlet carbonate and fracture carbonate. Then Sedimentary Carb, and include below these samples, and finally microbial. Also another extra group for the extra samples added for comparison. I think this would make it easier to identify the distribution of each carbonate factory in the plot rapidly.*

R: Thank you for your kind suggestion. It looks better than the original one. We have made this correction following your advice.

[revised manuscript text omitted]